# PROTOTYPICAL EXAMPLES IN DEEP LEARNING: METRICS, CHARACTERISTICS, AND UTILITY

## ABSTRACT

Machine learning (ML) research has investigated *prototypes*: examples that are representative of the behavior to be learned. We systematically evaluate five methods for identifying prototypes, both ones previously introduced as well as new ones we propose, finding all of them to provide meaningful but different interpretations. Through a human study, we confirm that all five metrics are well matched to human intuition. Examining cases where the metrics disagree offers an informative perspective on the properties of data and algorithms used in learning, with implications for data-corpus construction, efficiency, adversarial robustness, interpretability, and other ML aspects. In particular, we confirm that the "train on hard" curriculum approach can improve accuracy on many datasets and tasks, but that it is strictly worse when there are many mislabeled or ambiguous examples.

## 1 INTRODUCTION

When reasoning about ML tasks, it is natural to look for a set of training or test examples that is somehow *prototypical*—i.e., that is representative of the desired learned behavior. Although such prototypical examples have been central to several research efforts, e.g., in interpretability (Bien & Tibshirani, 2011) and curriculum learning (Bengio et al., 2009), no generally-agreed-upon definition seems to exist for prototypes, or their characteristics. For modern deep-learning models, whose behavior is often inscrutable, even the very existence and usefulness of prototypical examples has seemed uncertain until the recent work of Stock & Cisse (2017).

Inspired by that work we (1) identify a set of desirable properties for prototypicality definitions; (2) systematically explore different metrics used in prior work, as well as new metrics we develop, for identifying prototypical examples in both training and test data; (3) study the characteristics of those metrics' prototypes and their complement set—the *outliers*—using both quantitative measures and a qualitative human study; and, (4) evaluate the usefulness of prototypes for machine-learning purposes such as reducing sample complexity or improving adversarial robustness and interpretability.

Our prototypicality metrics are based on adversarial robustness, retraining stability, ensemble agreement, and differentially-private learning. As an independent result, we show that predictive stability under retraining strongly correlates with adversarial distance, and may be used as an approximation.

Unequivocally, we find that distinct sets of prototypical and outlier examples exist for the datasets we consider: MNIST (LeCun et al., 2010), Fashion-MNIST (Xiao et al., 2017), CIFAR-10 (Krizhevsky & Hinton, 2009), and ImageNet (Russakovsky et al., 2015). Between all of our metrics, as well as human evaluators, there is overall agreement on the examples that are prototypes and those that are outliers. Furthermore, the differences between metrics constitute informative exceptions, e.g., identifying uncommon submodes in the data as well as spurious, ambiguous, or misleading examples.

Usefully, there are advantages to training models using only prototypical examples: the models learn much faster, their accuracy loss is not great and occurs almost entirely on outlier test examples, and the models are both easier to interpret and more adversarially robust. Conversely, at the same sample complexity, significantly higher overall accuracy can be achieved by training models exclusively on outliers—once erroneous and misleading examples have been eliminated from the dataset.

## 2 DEFINING AND IDENTIFYING PROTOTYPES

In designing a metric to identify prototypes, many approaches may seem intuitive. We identify the following properties as desirable for any prototypicality metric:

- **Independent of the learning task:** A prototypicality metric should be applicable to all types of machine-learning tasks, whether they are based upon unsupervised or supervised approaches, or whether they constitute classification tasks or generative tasks, etc.

- **Independent of the modeling approach:** A metric should identify the overall same prototypical examples regardless of the machine-learning paradigm, model architecture, capacity, or hyperparameters that are used to learn the task.

- **Aligned with human intuition:** At least for tasks on which humans do well, prototypical examples identified by a metric should strongly overlap with those identified by humans.

- **Covers all apparent data modes:** A metric should provide a balanced view of all modes of prototypical examples (e.g., even when multiple disparate modes have a single output classification label). In the presence of imbalance in the frequency of data modes, the metric should provide coverage while reflecting frequency: a mode supported by only a handful of examples need not be as prototypical as other modes in the same class.

- **Provides a stable ranking:** A metric should not only identify examples, but also rank them in terms of prototypicality, and this rank should be stable (i.e., have low variance) even though the metric is likely computed by a randomized procedure.

- **Applies to both training and test data:** A metric should allow both training examples and test examples to be ranked in terms of prototypicality.

- **Predicts test accuracy:** Models that are trained only on prototypical training examples should still achieve good accuracy on the prototypical test examples—at least when the training and test datasets are balanced and the models use machine-learning methods known to perform well overall. Other cases need not be so predictive (e.g., if training is done only on outliers); we explore this experimentally in Section 3 and Section 4.

Although metrics for prototypicality should generally satisfy the above properties, they need not do so perfectly. In fact, our experimental results show that the differences between metrics can be highly informative. In particular, as described in Section 3.3, such differences can improve interpretability and provide insights into model behavior that are not achievable by a single, overall accuracy number, e.g., by giving explanations of failures by example (Caruana et al., 1999).

### 2.1 METRICS FOR PROTOTYPICALITY

A number of metrics for identifying prototypes might satisfy the desirable properties we identified above, possibly even the early methods based on concept similarity introduced by Zhang (1992). In addition to the work presented below, a complete survey of related work is found in the Appendix.

To start with, consider two strawmen prototypicality metrics based on either learning order or gradient magnitude. A model may be expected to learn prototypes early in training because they are presumably more common and somehow "simpler" than all the edge-case outliers. Conversely, near the end of training, one might expect the relative magnitude of the gradient $\|\nabla_\theta \ell(f_\theta(x))\|$ to be small for prototypical examples $x$, as each such example should have very little to teach the model. From this, two metrics can be defined, which we evaluated in experiments that averaged hundreds of model training runs to minimize random-initialization and stochastic-learning effects. Unfortunately, both of these metrics exhibited both very high variance as well as apparently-low signal, defining prototypes sets that appeared random upon inspection and that did not satisfy our stated desirable property of predicting test accuracy. Thus, we do not present results for these two metrics.

Instead, we define and apply the following five metrics, each of which ranks examples by their relative ordinal number (i.e., position in the sorted order of the measured value):

**Adversarial Robustness (adv):** Prototypical examples should be more adversarially robust. As a measure of prototypicality, the distance to the decision boundary measured by an adversarial-example attack was recently proposed and utilized by Stock & Cisse (2017). Specifically, for an

example $x$, the measure finds the perturbation $\delta$ with minimal $\|\delta\|$ such that the original $x$ and the adversarial example $x + \delta$ are classified differently (Biggio et al., 2013; Szegedy et al., 2013).

To compare prototypicality, the work of Stock & Cisse (2017) that inspired our current work used a simple and efficient $\ell_\infty$-based adversarial-example attack based on an iterative gradient descent introduced by Kurakin et al. (2016). That attack procedure computes gradients to find directions that will increase the model's loss on the input within an $\ell_\infty$-norm ball. They define prototypicality as the number of gradient descent iterations necessary to change the class of the perturbed input.

Instead, the adv metric ranks by the $\ell_2$ norm (or faster, less accurate $\ell_\infty$ norm) of the minimal-found adversarial perturbation (Carlini & Wagner, 2017). This is generally more accurate at measuring the distance to the decision boundary, but comes at a performance cost (it is on average 10-100$\times$ slower).

**Holdout Retraining (ret):**   The intuition beind our ret metric is that a model should treat a prototypical example the same regardless of whether, or when, it was used in the training process.

Assume we are given a training dataset $\mathcal{X}$, a disjoint holdout dataset $\bar{\mathcal{X}}$, and an example $x \in \mathcal{X}$ for which to assess prototypicality. To begin, we train a model $f(\cdot)$ on the training data $\mathcal{X}$ to obtain model weights $\theta$. We train this model just as how we would typically do—i.e., with the same learning rate schedule, hyper-parameter settings, etc. Then, we fine-tune the weights of this first model $f_\theta(\cdot)$ on the held-out training data $\bar{\mathcal{X}}$ to obtain new weights $\bar{\theta}$. To perform this fine-tuning, we use a smaller learning rate and train until the training loss stops decreasing. (We have found it is important to obtain $\bar{\theta}$ by fine-tuning $\theta$ as opposed to training from scratch; otherwise, the randomness of training leads to unstable rankings that yield specious results.) Finally, given these two models, we compute the prototypicality of $x$ as the difference $\|f_\theta(x) - f_{\bar{\theta}}(x)\|$. The exact choice of metric $\|\cdot\|$ is not important; the results in this paper use the symmetric KL-divergence.

While this metric is similar to the one considered in Ren et al. (2018), it differs in important ways: notably, our holdout retraining metric is conceptually simpler, more stable numerically, and more computationally efficient (because it does not require a backward pass to estimate gradients in addition to the forward pass needed to compare model outputs). Because our metric is only meaningful for data used to train the initial model, in order to measure the prototypicality of arbitrary test points, we actually *train on the test data* and perform holdout retraining on the original training data.

**Ensemble Agreement (agr):**   Prototypical examples should be easy for many types of models to learn. We train multiple models of varying capacity on different subsets of the training data (see Appendix C). The agr metric ranks examples' prototypicality based on the agreement within this ensemble, as measured by the symmetric KL-divergence between the models' output. Concretely, we train many models $f_{\theta_i}(\cdot)$ and, for each example $x$, evaluate the model predictions, and then compute $\frac{1}{N^2} \sum_{i=1}^{N} \sum_{j=1}^{N} \text{JS-Divergence}(f_{\theta_i}(x), f_{\theta_j}(x))$ to rank the example's prototypicality.

**Model Confidence (conf):**   We expect models to be confident on prototypical examples. Based on an ensemble of models like that used by the agr metric, the conf metric ranks examples by the mean confidence in the models' predictions, i.e., ranking each example $x$ by $\frac{1}{N} \sum_{i=1}^{N} \max f_{\theta_i}(x)$.

**Privacy-preserving Training (priv):**   We can expect prototypical, well-represented modes of examples to be classified properly by models (e.g., deep neural networks) even when trained with guarantees of differential privacy (Abadi et al., 2016; Papernot et al., 2016). However, such privacy-preserving models should exhibit significantly reduced accuracy on any rare or exceptional examples, because differentially-private learning attenuates gradients and introduces noise to prevent the details about any specific training examples from being memorized. Outliers are disproportionally likely to be impacted by this attenuation and added noise, whereas the common signal found across many prototypes must have been preserved in models trained to reasonable accuracy.

Our priv metric is based on training an ensemble of models with increasingly greater $\varepsilon$ privacy (i.e., more attenuation and noise) using $\varepsilon$-differentially-private stochastic gradient descent (Abadi et al., 2016). Our metric then ranks the prototypicality of an example based on the maximum $\varepsilon$ privacy at which the example is correctly classified in a reliable manner (which we take as being also classified correctly in 90% of less-private models). This ranking embodies the intuition that the more tolerant an example is to noise and attenuation during learning, the more prototypical it is.

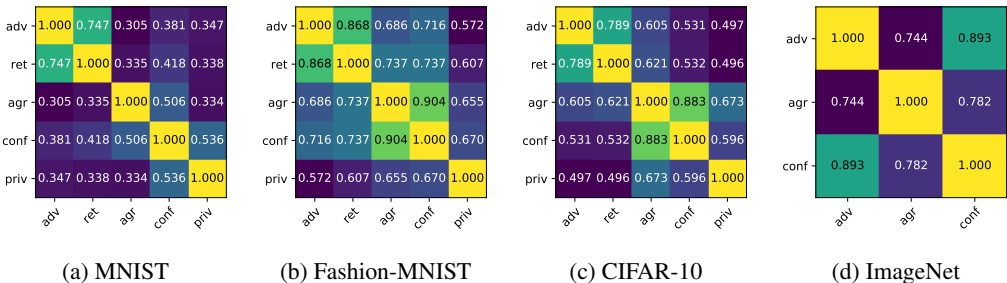

|          (a) MNIST          |          (b) Fashion-MNIST          |          (c) CIFAR-10          |          (d) ImageNet          |

Figure 1: Correlation coefficients for our five prototypicality metrics on four common datasets.

# 3 EVALUATING THE FIVE DIFFERENT PROTOTYPICALITY METRICS

As the first step in an evaluation, it is natural to consider how the above five metrics satisfy the properties we identified earlier—in Section 2—as being desirable for all prototypicality metrics.

In general, all our five metrics satisfy these desirable properties. All five metrics can be applied to both training and test data to induce a ranking and, empirically, we find this ranking is stable when computed multiple times. Furthermore, overall, each of the five metrics exhibits good coverage, with a few informative exceptions, and provides a view of all distinct, prototypical data modes that is proportionally balanced—even in the presence of data skew—in particular across output class labels. Notably, none of the metrics fails by ranking some class of labeled examples (e.g., the easiest-to-learn class) as being strictly more prototypical than all examples with other labels; even within each class, any substantial fraction of the most prototypical examples exhibits good modal coverage. Our metrics are widely applicable, as they are not specific to any learning task or model (some, like ret and priv might be applicable even to unsupervised learning), and experimentally we have confirmed that the metrics give overall the same results despite large changes in hyperparameters or even the model architecture. Finally, as described further in Section 4, all of our metrics provide strong predictive accuracy: training on prototypes gives good test performance on prototypes. (Experimental results supporting the above observations can be seen in the Appendices.)

## 3.1 QUANTITATIVE EVALUATION OF CORRELATION COEFFICIENTS

Figure 1 shows the correlation coefficients computed pairwise between each of our metrics for all our datasets, as well as on three metrics for ImageNet[1] (the tables are symmetric across the diagonal). The metrics are overall strongly correlated, and the differences in correlation are informative. Unsurprisingly, since they measure very similar properties, the agr (ensemble agreement) and the conf (model confidence) show the highest correlation, However, somewhat unexpectedly, we find that the adv (adversarial robustness) correlates very strongly with ret (retraining distance). This is presumably because these two metrics both measure the distance to a model's decision boundary—even though adv measures this distance by perturbing each example while ret measures how the evaluation of each example is affected when models' decision boundaries themselves are perturbed.

(This strong correlation between adv and ret is a new result that may be of independent interest and some significance. Measurement of adversarial distance is a useful and highly-utalized technique, but it is undefined or ill-defined on many learning tasks and its computation is difficult, expensive, and hard to calibrate. On the other hand, given any holdout dataset and any measure of divergence, the ret metric we define in Section 2.1 should be easily computable for any ML model or task.)

## 3.2 QUALITATIVE EVALUATION BY INFORMAL INSPECTION AND BY A HUMAN STUDY

It remains to be established that our five metrics rank examples in a manner that corresponds to human intuition. For this, we perform a subjective visual inspection of how the different metrics rank the example training and test data for different machine-learning tasks. We establish that there is a clear, intuitive difference between the prototype and outlier extremes of the ranking. Informally,

---

[1]Computational constraints prevented us from completing experiments on the 1.2M ImageNet examples as our ret and priv metrics are resource demanding. The full data will be included in a revision of this paper.

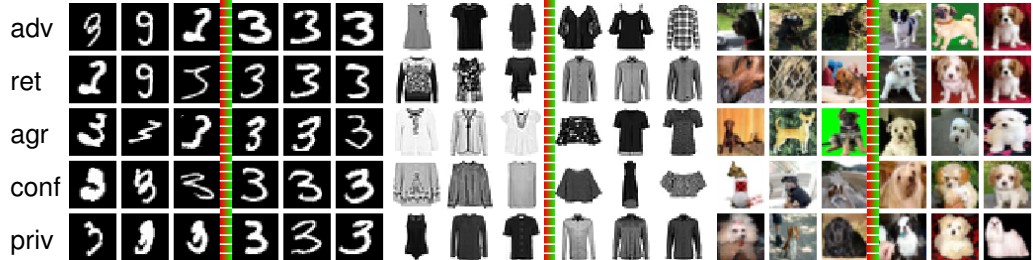

Figure 2: Our five metrics' least and most prototypical training examples, separated by a red/green bar, for three classes of MNIST, Fashion-MNIST, and CFIAR-10 (all classes shown in Appendix C). The separation is clearly informative e.g., revealing an outlier "9" mislabeled as a three; also, looking at the conf prototypes reveals an atypical dress-like "shirt" that was memorized during training.

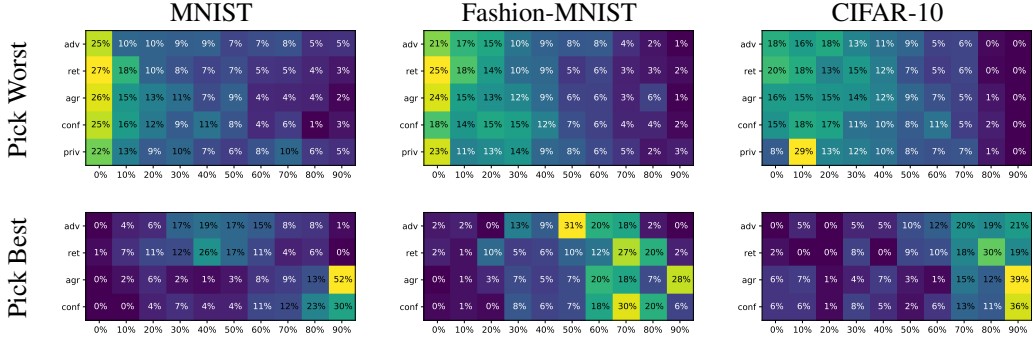

Table 1: Results of a human study of Mechancial Turk workers selecting the best or worst example among a random collection of 9 training-data images. For each prototypicality metric, the tables show what percent of workers selected examples in each 10% split of the metric's sorted ranking (e.g., 52% of the MNIST images picked as the best one rank in the 90[th] percentile on the agr metric).

Figure 2 and the figures in Appendix C confirm that there is an obviously apparent difference between at least the extreme outliers and prototypes in the MNIST, Fashion-MNIST, and CIFAR-10 training examples, and the ImageNet validation examples—although between datasets and classes, the five metrics differ in how clearly this difference can be seen.

To validate and quantify how our metrics correlate with human perception, we performed an online human study using Amazon's Mechanical Turk service. For each output class in the training and test data of MNIST, Fashion-MNIST, and CIFAR-10, the study had human evaluators choose the image that was most or least representative of the class from amongst 9 randomly-selected images. In the study, over 100,000 images were assessed by over 400 different human evaluators.

Concretely, in this study (as shown in Appendix E), each human evaluator saw a 3x3 grid of 9 random images and was asked to pick the worst image—or the best image—and this was repeated multiple times. Evaluators exclusively picked either best or worst images and were only shown random images from one output class under a heading with the label name of that class; thus one person would pick only the best MNIST digits "7" while another picked only the worst CIFAR-10 "cars." (As dictated by good study design, we inserted "Gold Standard" questions with known answers to catch workers answering randomly or incorrectly, eliminating the few such workers from our data.) For all datasets, picking non-representative images proved to be the easier task: in a side study where 50 evaluators were shown the same identical 3x3 grids, agreement was 80% on the worst image but only 27% on the best image (random choice would give 11% agreement).

The results of our human study are presented in Table 1. One of the takeaways is that the assessment of human evaluators is strongly correlated with each one of our metrics: humans mostly picked low-prototypicality images as the worst examples and examples with significantly higher-prototypicality as being the best. Somewhat surprisingly, there are some large, notable differences between metrics and datasets in their correspondence to human perception—e.g., for Pick Worst, the lowest percentile split of the priv metric does poorly (at 8%) whereas the next does extremely well (at 29%)—suggesting that further investigation is warranted.

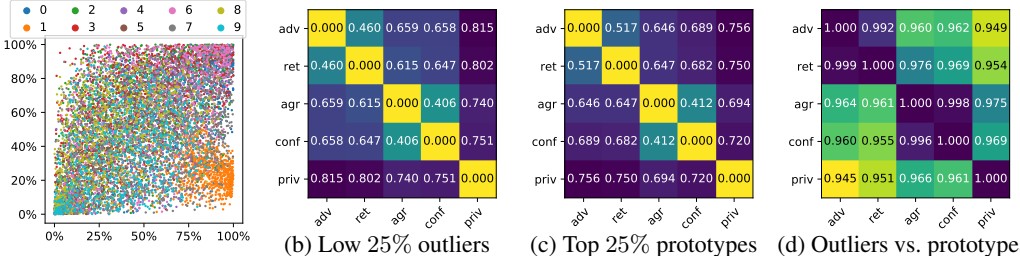

(a) Scatter plot comparing the adv vs. conf ranks.

(b) Low 25% outliers    (c) Top 25% prototypes    (d) Outliers vs. prototypes

(b-d) The Jaccard distance between the sets of the 25% most lowest-ranked outliers and the 25% most highest-ranked prototypes for each class on Fashion-MNIST.

Figure 3: Comparing the differences between the metrics on (a) MNIST and (b-d) Fashion-MNIST.

## 3.3 Comparing Prototypicality Metrics and their Characteristics

Because our five metrics for prototypicality are not perfectly correlated, there are likely to be many examples that are highly prototypical under one metric but not under another. To quantify the number and types of those differences we can try looking at their visual correlation in a scatter plot; doing so can be informative, as can be seen in Figure 3(a) where the easily-learned, yet fragile, examples of class "1" in MNIST models have high confidence but low adversarial robustness. To further quantify, we can also compute the Jaccard distance to assess the relative size of the intersections between different sets of prototypes and outliers; the rest of Figure 3 shows the results of doing so at the 25% threshold for Fashion-MNIST. The results show substantial disagreement between metrics.

To understand disagreements, we can consider examples that are prototypical in one metric but outliers in others, first combining the union of adv and ret prototypes into a single boundary metric, and the union of adv and ret prototypes into an ensemble metric, because of their high correlation.

**Memorized exceptions:** Recalling the unusual dress-looking "shirt" of Figure 2, and how it seemed to have been memorized with high confidence, we can intersect the top 25% prototypical ensemble images with the bottom-half outliers in both the boundary and priv metrics. For the Fashion-MNIST "shirt" class, this set—visually shown in Figure 4 on the right—includes not only the dress-looking example but a number of other atypical "shirt" images, including some looking like shorts. Also apparent in the set are a number of T-shirt-like and pullover-like images, which are misleading, given the other output classes of Fashion-MNIST. For these sets, which are likely to include spurious, erroneously-labeled, and inherently ambiguous examples, we

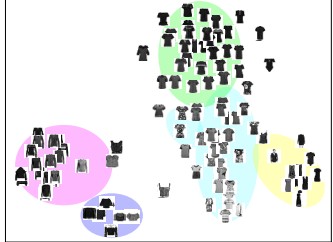

Figure 4: Exceptional "shirts."

use the name *memorized exceptions* because they must be memorized as exceptions for models to have been able to reach very high confidence during training. Similarly, Figure 5(a) shows a large (green) cluster of highly ambiguous boot-like sneakers, which appear indistinguishable from a cluster of memorized exceptions in the Fashion-MNIST "ankle boot" class (see Appendix C).

**Uncommon submodes:** On the other hand, the priv metric is based on differentially-private learning which ensures that no small group of examples can possibly be memorized: the privacy stems from adding noise and attenuating gradients in a manner that will mask the signal from rare examples during training. This suggests that we can find *uncommon submodes* of the examples in learning tasks by intersecting the bottom-most outlier examples on the priv metric with the union of top prototypes in the boundary and ensamble metrics. Figure 5(b) shows uncommon submodes discovered in MNIST using the 25% lowest outliers on priv and top 50% prototypes on other metrics. Notably, all of the "serif 1s" in the entire MNIST training set are found as a submode.

**Canonical prototypes:** Finally, we can simply consider the intersection of the sets of all the topmost prototypical examples in all of our metrics. The differences between our metrics should ensure that this intersection is free of spurious or misleading examples; yet, our experiments and human study suggest the set will provide good coverage. Hence, we call this set *canonical prototypes*. Figure 5(c) shows the airplanes that are canonical prototypes in CIFAR-10.

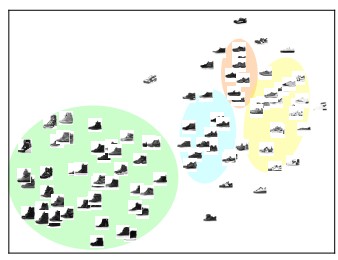 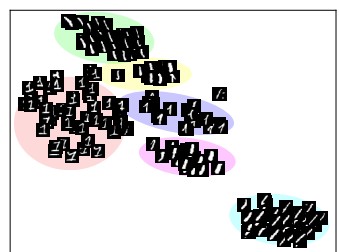 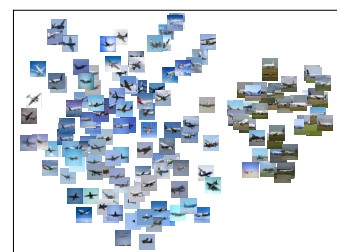

(a) Memorized exceptions in the Fashion-MNIST "sneaker" class.

(b) Uncommon submodes found within the MNIST "1" class.

(c) Canonical prototypes in the CIFAR-10 "airplane" class.

Figure 5: Our metrics' prototype and outlier sets reveal interesting examples, which can be clustered.

To further aid interpretability in Figures 4 and 5, we perform a combination of dimensionality reduction and clustering. Concretely, we apply t-SNE (Maaten & Hinton, 2008) on the pixel space (for MNIST and Fashion-MNIST) or ResNetv2 feature space (for CIFAR10) to project the example sets into two dimensions. We then cluster this two-dimensional data using HDBSCAN (Campello et al., 2013), a hierarchical and density-based clustering algorithm which does not try to assign all points to clusters—which not only can improve clusters but also identify spurious data. We believe that other types of data projection and clustering could also be usefully applied to our metrics, and offer significant insight into ML datasets. (See Appendix G for this section's figures shown larger.)

## 4 UTILIZING PROTOTYPES TO IMPROVE ASPECTS OF MACHINE LEARNING

By using prototype metrics, we can improve models' sample complexity, accuracy, or robustness.

We perform two experiments on the three datasets to investigate whether it is better to train on outliers or on prototypes—exploring the "train on hard data" vs. "train on easy data" question of curriculum learning (Ren et al., 2018), which is discussed in Appendix A. To begin, we order all training data according to its prototypicality as measured by our adv metric.[2]

First, we experiment with training on splits of $5,000$ training examples (approximately $10\%$ of the training data) chosen by taking the $k$-th most prototypical example to the $(k+5000)$-th most prototypical. As shown in Figure 6, we find that the split that yields the most accurate model varies substantially across the datasets and tasks. On MNIST, training on the least prototypical examples gives the highest accuracy; conversely, on CIFAR-10, training on nearly-the-most prototypical examples gives the highest accuracy. We conjecture this is due to the dataset complexity: because nearly all of MNIST is very easy, it makes sense to train on the hardest, most outlier examples. However, because CIFAR-10 is very difficult, training on very prototypical examples is better.

Notably, many of the CIFAR-10 and Fashion-MNIST outliers appear to be inherently misleading or ambiguous examples, and several are simply erroneously labeled. We find that about $10\%$ of the first 5,000 outliers meet our definition of memorized exceptions. Also, we find that inserting 10% label noise causes model accuracy to decrease by about $10\%$, regardless of the split trained on— i.e., that to achieve high accuracy on small training data erroneous and misleading outliers must be removed—and explaining the low accuracy shown on the left in the graph of Figures 6(b) and 6(c).

The prior experiment assumes the amount of data is fixed, and we must choose which percentile of data to use. Now, we examine what the best strategy is to apply if the amount of training data is not fixed, and ask: is it better to train on the $k$-most or $k$-least prototypical examples? Again, we find the answer depends on the dataset. On MNIST, training on the $k$-least prototypical examples is always better. However, on Fashion-MNIST and CIFAR-10, training on the prototypical examples is better when $k$ is small, but as soon as we begin to collect more than roughly $10,000$ examples for Fashion-MNIST or $20,000$ for CIFAR-10, training on the outliers begins to give more accurate models. However, we find that training only on the most prototypical examples found in the training data gives extremely high test accuracy on the prototypical examples found in the test data.

---

[2]We use the adv metric since it is well-correlated to human perception and unrelated to model performance.

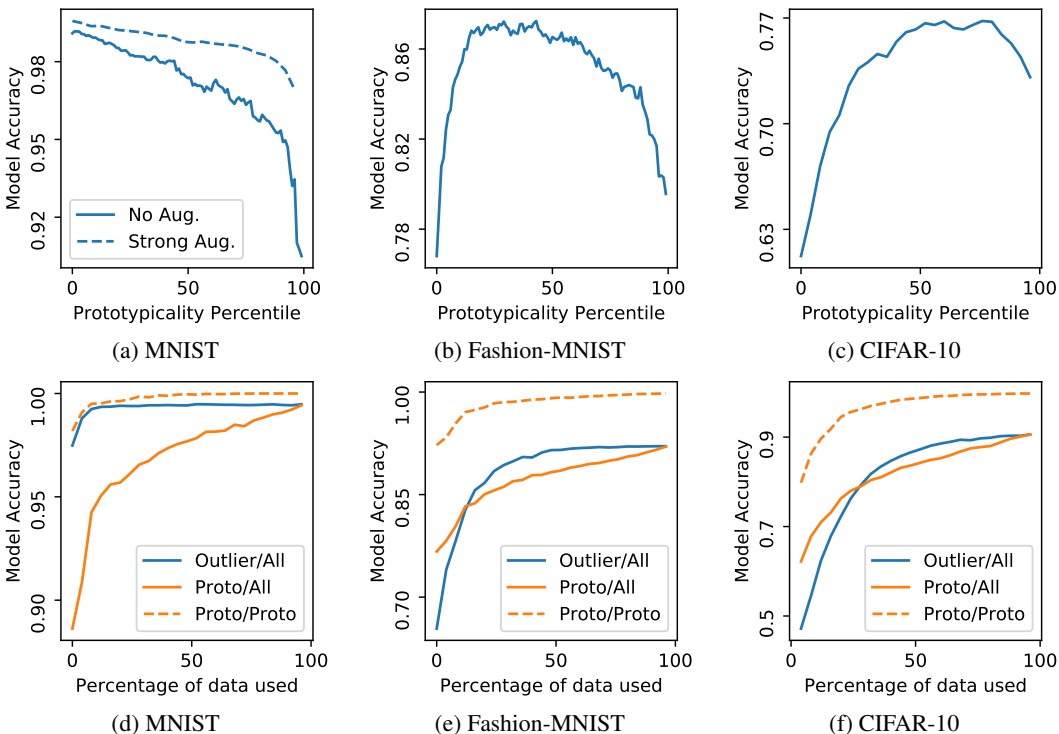

Figure 6: Comparing training on the $k$-most prototypical and $k$-least prototypical examples (as determined by the adv metric). The first row of subfigures plots the model accuracy resulting from training on a slice of $5,000$ training examples consecutively ranked by prototypicality; in each prototypicality increases from left to right (i.e., most extreme outliers are on the far left, most protypical training examples are on the far right). On MNIST (a) training on the outliers gives strictly higher accuracy. However, on Fashion-MNIST (b) and CIFAR-10 (c), given only a limited amount of training data, prototype-training is better; but if more data is available outlier-training becomes superior. At least for some datasets, heavy data augmentation and regularization techniques like dropout can partially recover the utility loss that results from training only with more protoypical examples; the dotted line in subfigure (a) shows the result of one such experiment for MNIST.

The second row of subfigures (d), (e), and (f) plots the model accuracy resulting from training on a fractional slice of prototypicality-ranked training examples, with the fraction increased from small percentages (on the left) to the entire training data (extreme right). One solid line (blue) shows the accuracy resulting from training on training data fractions that are becoming increasingly more prototypical (i.e., the left-most part of the blue line results from training on the most outlier fraction of training examples); conversely, the other solid line (yellow) shows accuracy from training on an increasing, less-and-less prototypical fraction of training data, from left to right. Finally, the dotted line (yellow) in subfigures (d), (e), and (f) shows the models' test accuracy on only those examples in the test data that have been deemed prototypical; as can be seen, even training only on a small fraction of the most prototypical training examples can suffice to achieve high accuracy on prototypical test examples.

### 4.1 TRAINING ON PROTOTYPES GIVES SIMPLER DECISION BOUNDARIES

While training exclusively on prototypes often gives inferior accuracy compared to training on the outliers, the former has the benefit of obtaining models with simpler decision boundaries. Thus, it is natural to ask whether training on prototypes gives models that are more robust to adversarial examples. In fact, prior work has found that discarding outliers from the training data can help with both classifying and detecting adversarial examples (Liu et al., 2018).

To show that simpler boundaries can lead to more robustness, we train models on fixed-sized subsets of the data where we vary the prototypicality of the $5,000$ training points included in the subset. For each model, we then compute the mean $\ell_\infty$ adversarial distance needed to find adversarial examples. As shown in Figure 10 of Appendix F, the Fashion-MNIST and CIFAR-10 models that are trained on prototypical examples are *more* robust to adversarial examples than those trained on a slice of training data that is mostly made up of outliers. However, these models trained on a slice of $5,000$ prototypical examples remain comparably robust to a baseline model trained on the entire data.

## 5 CONCLUSION

This paper explores prototypes: starting with the properties we would like them to satisfy, then evaluating metrics for computing them, and discussing how we can utilize them during training. The five metrics we study all are highly correlated, and capture human intuition behind what is meant by "prototypical". When the metrics disagree on the prototypicality of an example, we can often learn something interesting about that example (e.g., that it is from a rare submode of a class). Further, we explore the many reasons to utalize prototypes: we find that models trained on prototypes often have simpler decision boundaries and are thus more adversarially robust. However, training only on prototypes often yields inferior accuracy compared to training on outliers. We believe that further exploring metrics for identifying prototypes and developing methods for using them during training is an important area of future work, and hope that our analysis will be useful towards that end goal.

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

## APPENDIX

## A   RELATED WORK

**Prototypes.**   At least since the work of Zhang (1992) which was based on intra- and inter-concept similarity, prototypes have been examined using several metrics derived from the intuitive notion that one could find "quintessential observations that best represent clusters in a dataset" (Kim et al., 2014). Several more formal variants of this definition were proposed in the literature—along with corresponding techniques for finding prototypes. Kim et al. (2016) select prototypes according to their maximum mean discrepancy with the data, which assumes the existence of an appropriate kernel for the data of interest. Li et al. (2017) circumvent this limitation by prepending classifiers with an autoencoder projecting the input data on a manifold of reduced dimensionality. A prototype layer, which serves as the classifier's input, is then trained to minimize the distance between inputs and a set of prototypes on this manifold. While this method improves interpretability by ensuring that prototypes are central to the classifier's logic, it does require that one modify the model's architecture. Instead, metrics considered in our manuscript all operate on existing architectures. Stock & Cisse (2017) proposed to use distance to the boundary—approximately measured with an adversarial example algorithm—as a proxy for prototypicality.

**Other interpretability approaches.**   Prototypes enable interpretability because they provide a subset of examples that summarize the original dataset and best explain a particular decision made at test time (Bien & Tibshirani, 2011). Other approaches like saliency maps instead synthetize new inputs to visualize what a neural network has learned. This is typically done by gradient descent with respect to the input space (Zeiler & Fergus, 2014; Simonyan et al., 2013). Because they rely on model gradients, saliency maps can be fragile and only locally applicable (Fong & Vedaldi, 2017).

Beyond interpretability, prototypes are also motivated by additional use cases, some of which we discussed in Section 4. Next, we review related work in two of these applications: namely, curriculum learning and reducing sample complexity.

**Curriculum learning.**   Based on the observation that the order in which training data is presented to the model can improve performance (e.g., convergence) of optimization during learning and circumvent limitations of the dataset (e.g., data imbalance or noisy labels), curriculum learning seeks to find the best order in which to analyze training data (Bengio et al., 2009). This first effort further hypothesizes that easy-to-classify samples should be presented early in training while complex samples gradually inserted as learning progresses. While Bengio et al. (2009) assumed the existence of hard-coded curriculum labels in the dataset, Chin & Liang (2017) sample an order for the training set by assigning each point a sampling probability proportional to its leverage score—the distance between the point and a linear model fitted to the whole data. Instead, we use metrics that also apply to data that cannot be modelled linearly.

The curriculum may also be generated online during training, so as to take into account progress made by the learner (Kumar et al., 2010). For instance, Katharopoulos & Fleuret (2017) train an auxiliary LSTM model to predict the loss of training samples, which they use to sample a subset of training points analyzed by the learner at each training iteration. Similarly, Jiang et al. (2017) have an auxiliary model predict the curriculum. This auxiliary model is trained using the learner's current feature representation of a smaller holdout set of data for which ground-truth curriculum is known.

However, as reported in our experiments, training on easy samples is beneficial when the dataset is noisy, whereas training on hard examples is on the contrary more effective when data is clean. These observations oppose self-paced learning (Kumar et al., 2010) with hard example mining (Shrivastava et al., 2016). Several strategies have been proposed to perform better in both settings. Assuming the existence of a holdout set as well, Ren et al. (2018) assign a weight to each training example that characterizes the alignment of both the logits and gradients of the learner on training and heldout data. Chang et al. (2017) propose to train on points with high prediction variance or whose average prediction is close from the decision threshold. Both the variance and average are estimated by analyzing a sliding window of the history of prediction probabilities throughout training epochs.

**Sample complexity.** Prototypes of a given task share some intuition with the notion of core-sets (Agarwal et al., 2005; Huggins et al., 2016; Bachem et al., 2017; Tolochinsky & Feldman, 2018) because both prototypes and coresets describe the dataset in a more compact way—by returning a (potentially weighted) subset of the original dataset. For instance, clustering algorithms may rely on both prototypes (Biehl et al., 2016) or coresets (Biehl et al., 2016) to cope with the high dimensionality of a task. However, prototypes and coresets differ in essential ways. In particular, coresets are defined according to a metric of interest (e.g., the loss that one would like to minimize during training) whereas prototypes are independent of any machine-learning aspects as indicated in our list of desirable properties for prototypicality metrics from Section 2.

Taking a different approach, Wang et al. (2018) apply influence functions (Koh & Liang, 2017) to discard training inputs that do not affect learning. Conversely, for MNIST, we found in our experiments that removing individual training examples did not have a measurable impact on the predictions of individual test examples. Specifically, we trained many models to 100% training accuracy where we left one training example out for each model. There was no statistically significant difference between the models predictions on each individual test example

## B  FIGURES OF OUTLIERS AND PROTOTYPES

The following are training examples from MNIST, FashionMNIST, and CIFAR10 that are identified as most outlier (left of the red bar) or prototypical (right of the green bar). Images are presented in groups by class. Each row in these groups corresponds to one of the five metrics in Section 2.1.

### B.1  MNIST EXTREME OUTLIERS AND PROTOTYPES

All MNIST results were obtained with a CNN made up of two convolutional layers (each with kernel size of 5x5 and followed by a 2x2 max-pooling layer) and a fully-connected layer of 256 units. It was trained with Adam at a learning rate of $10^{-3}$ with a $10^{-3}$ decay. When an ensemble of models was needed (e.g., for the agr metric), these were obtained by using different random initializations.

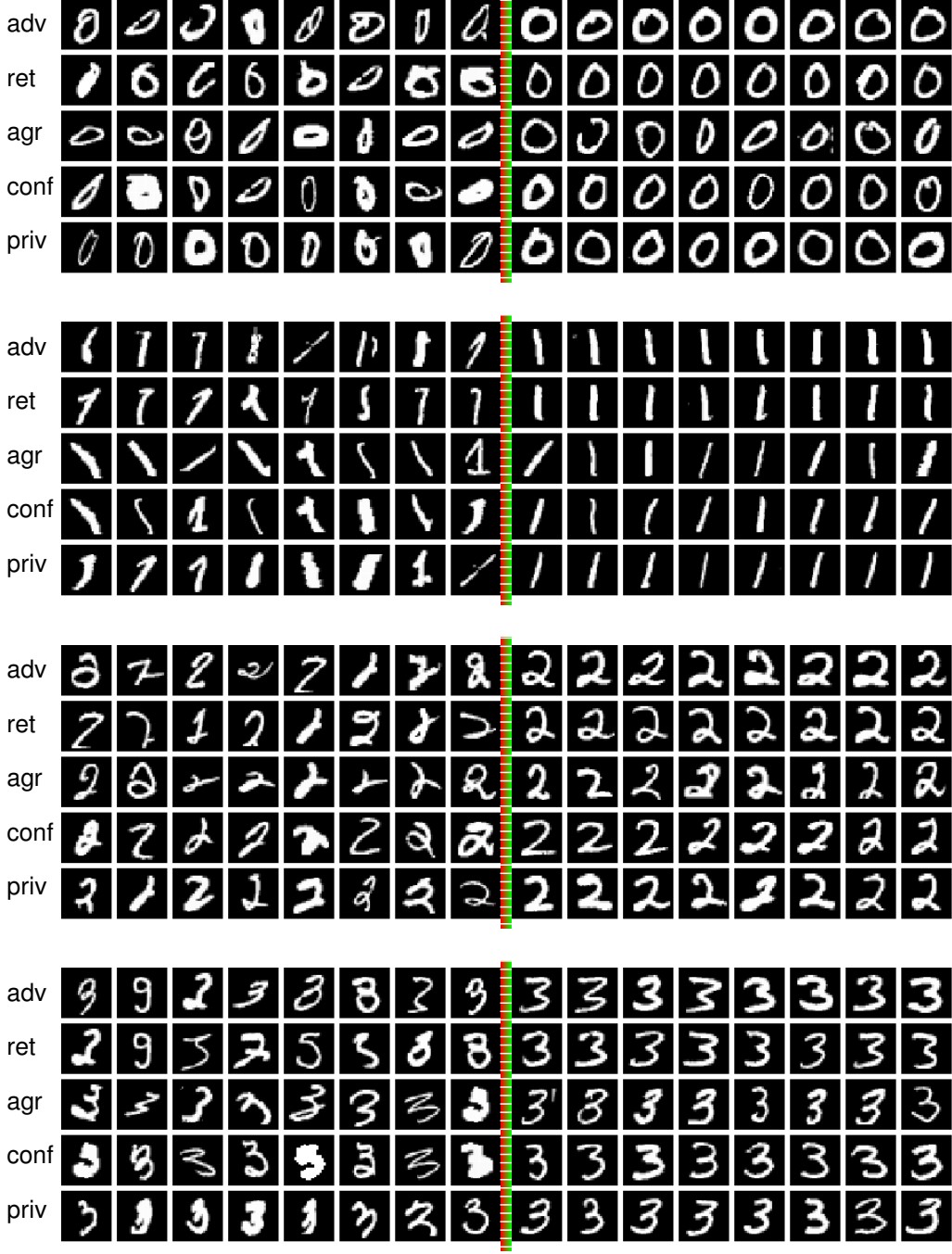

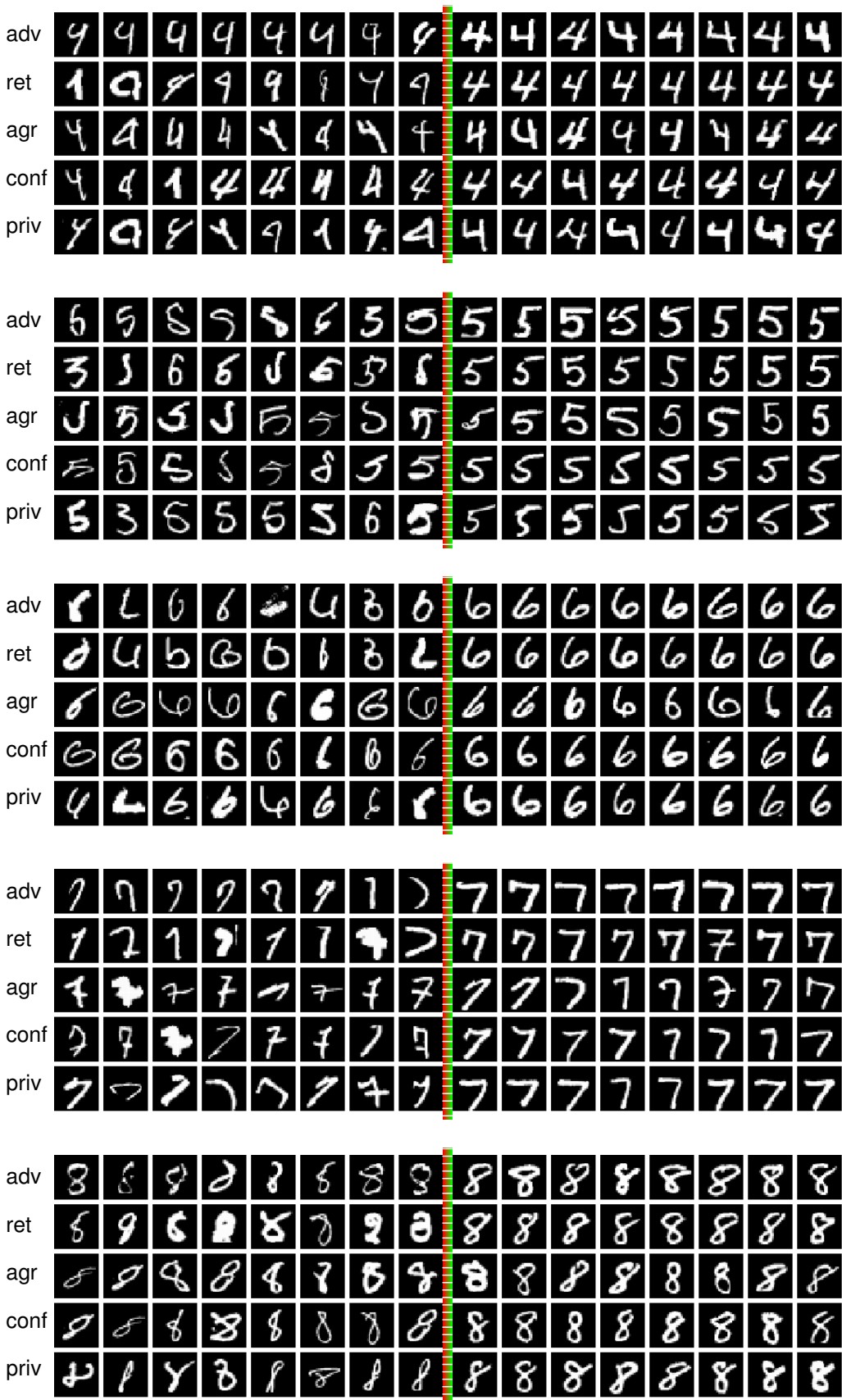

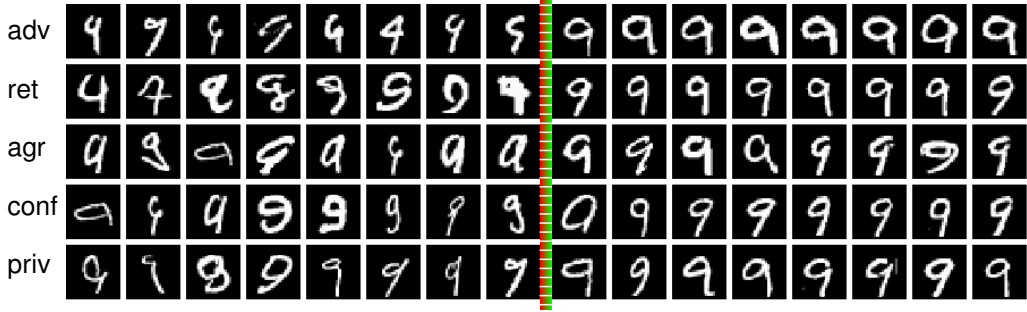

## B.2 FASHION-MNIST EXTREME OUTLIERS AND PROTOTYPES

The Fashion-MNIST model architecture is identical to the one used for MNIST. It was also trained with the same optimizer and hyper-parameters.

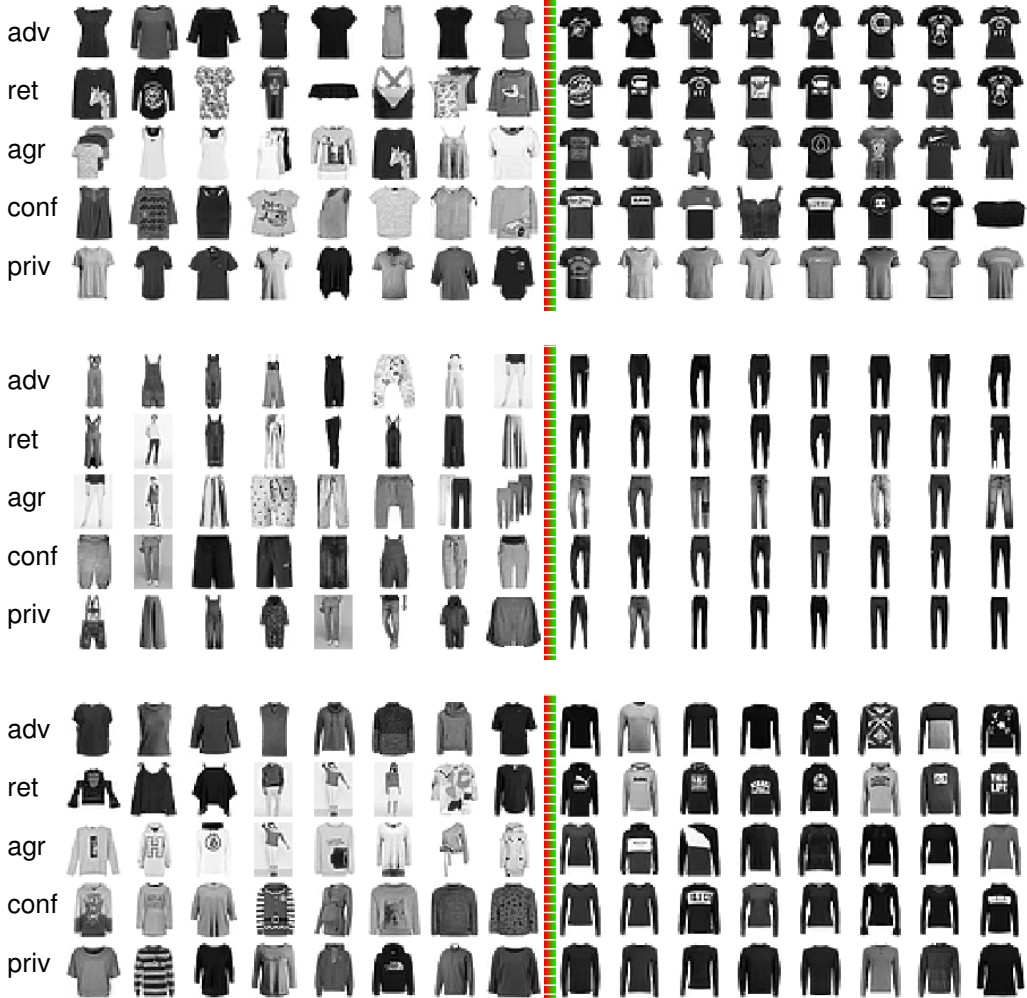

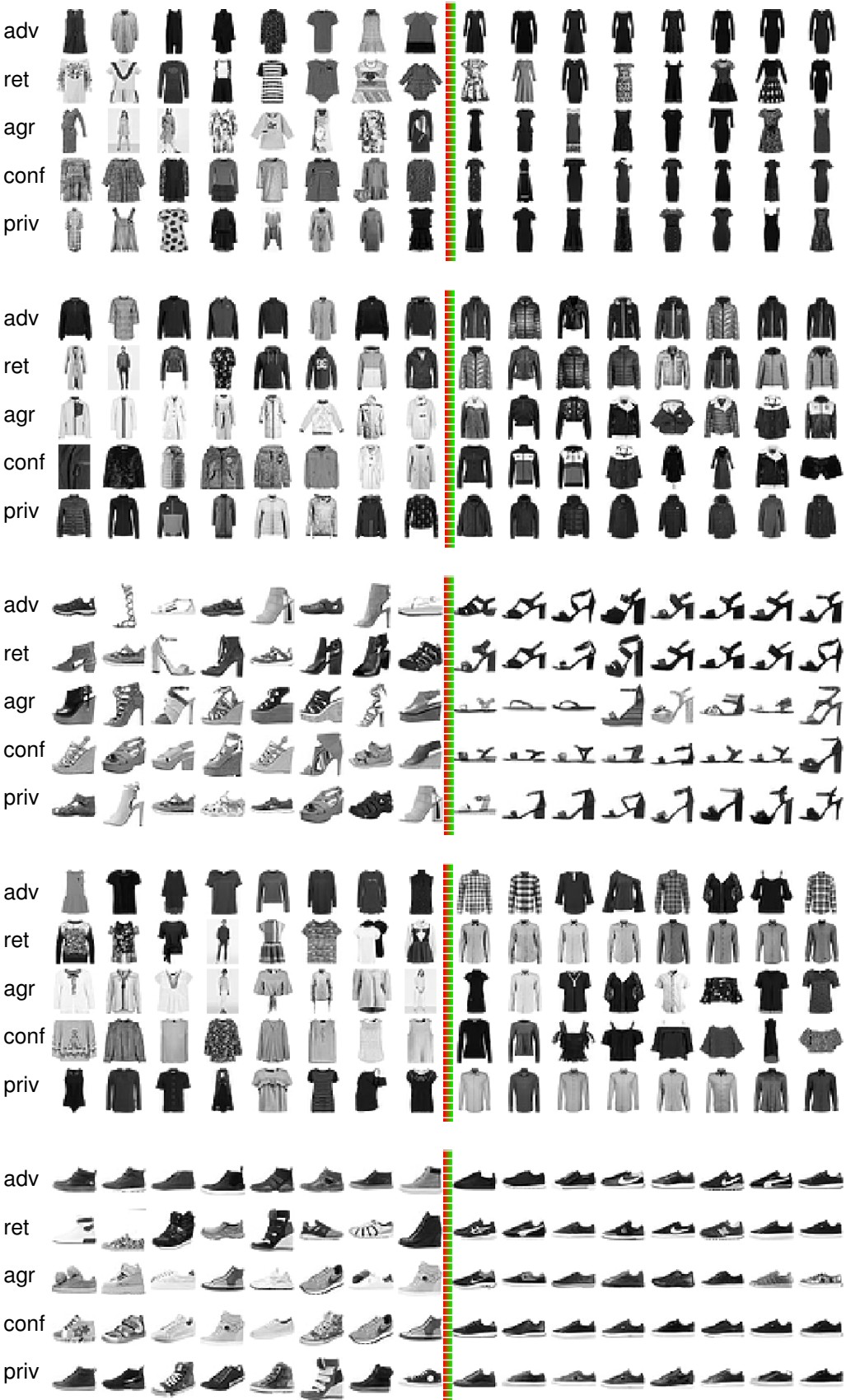

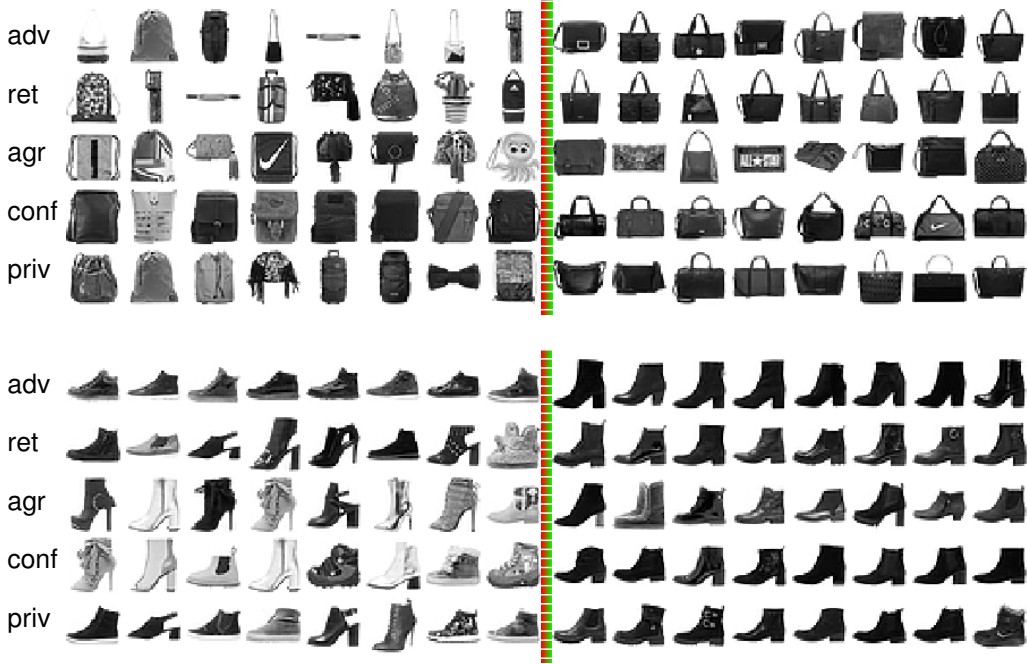

## B.3 CIFAR EXTREME OUTLIERS AND PROTOTYPES

All CIFAR results were obtained with a ResNetv2 trained on batches of 32 points with the Adam optimizer for 100 epochs at an initial learning rate of $10^{-3}$ decayed down to $10^{-4}$ after 80 epochs. We adapted the following data augmentation and training script: https://raw.githubusercontent.com/keras-team/keras/master/examples/cifar10_resnet.py When an ensemble of models was needed (e.g., for the agr metric), these were obtained by using different random initializations.

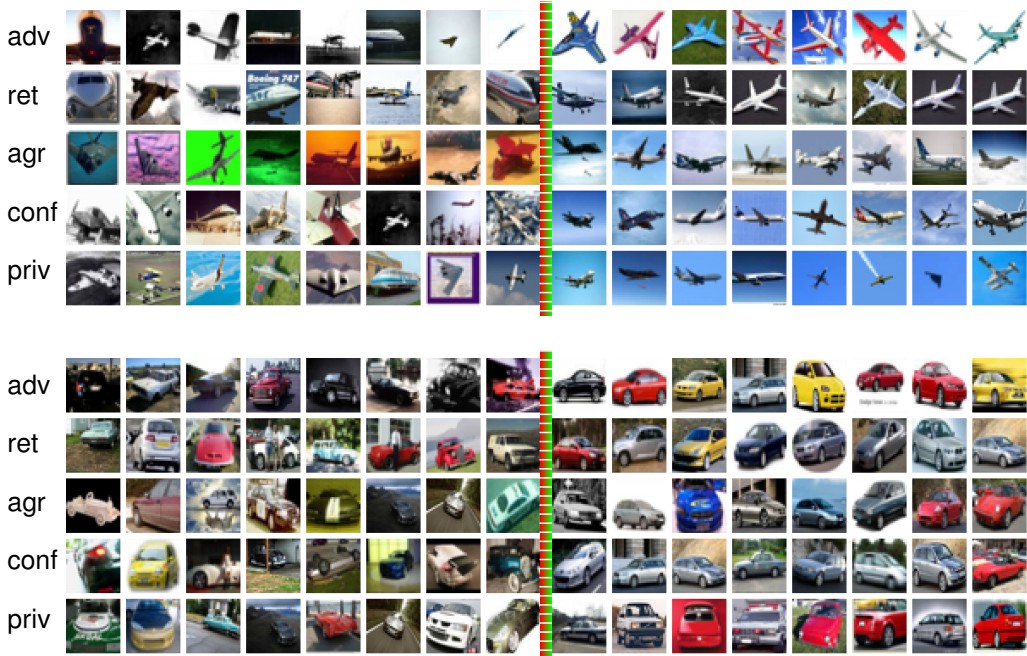

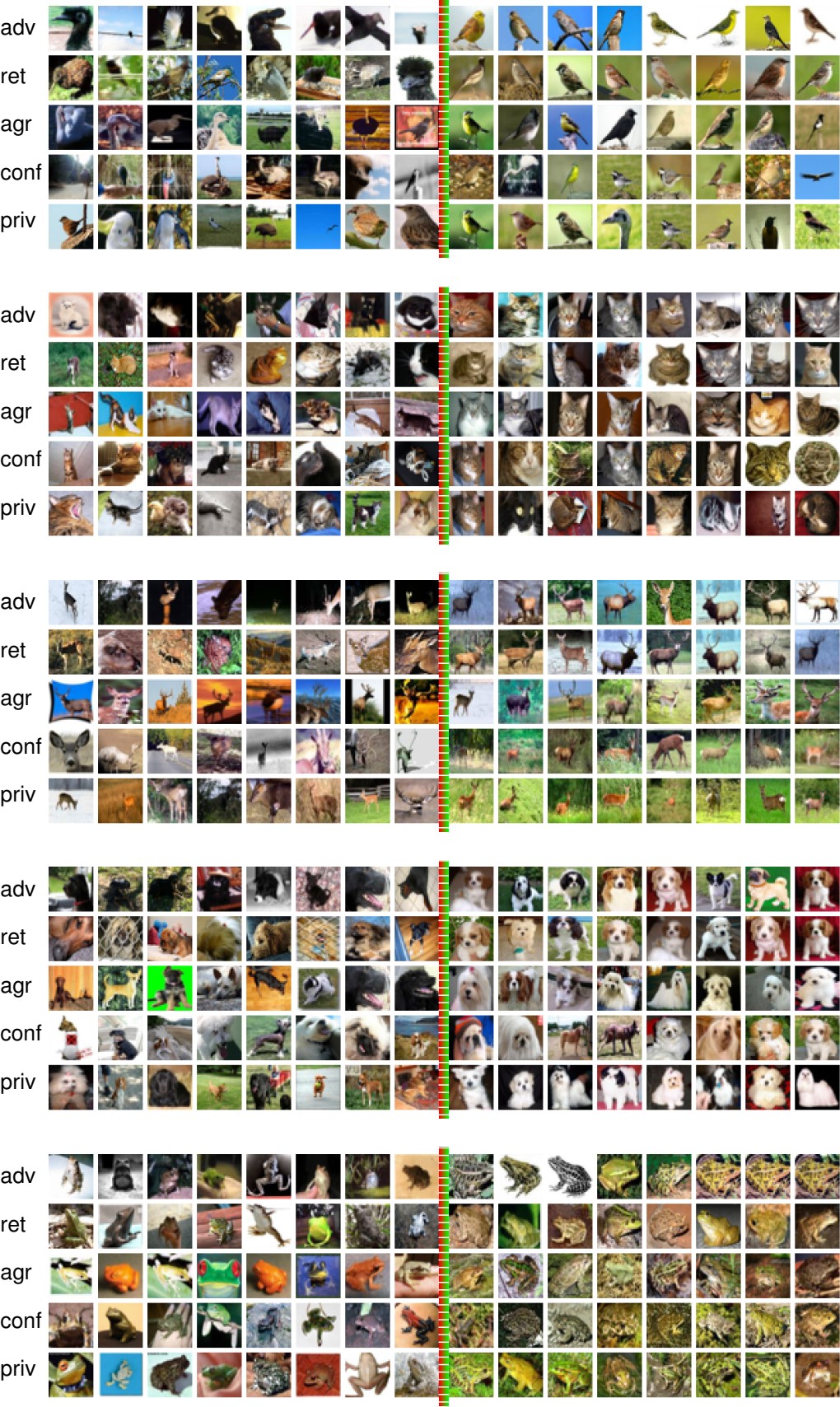

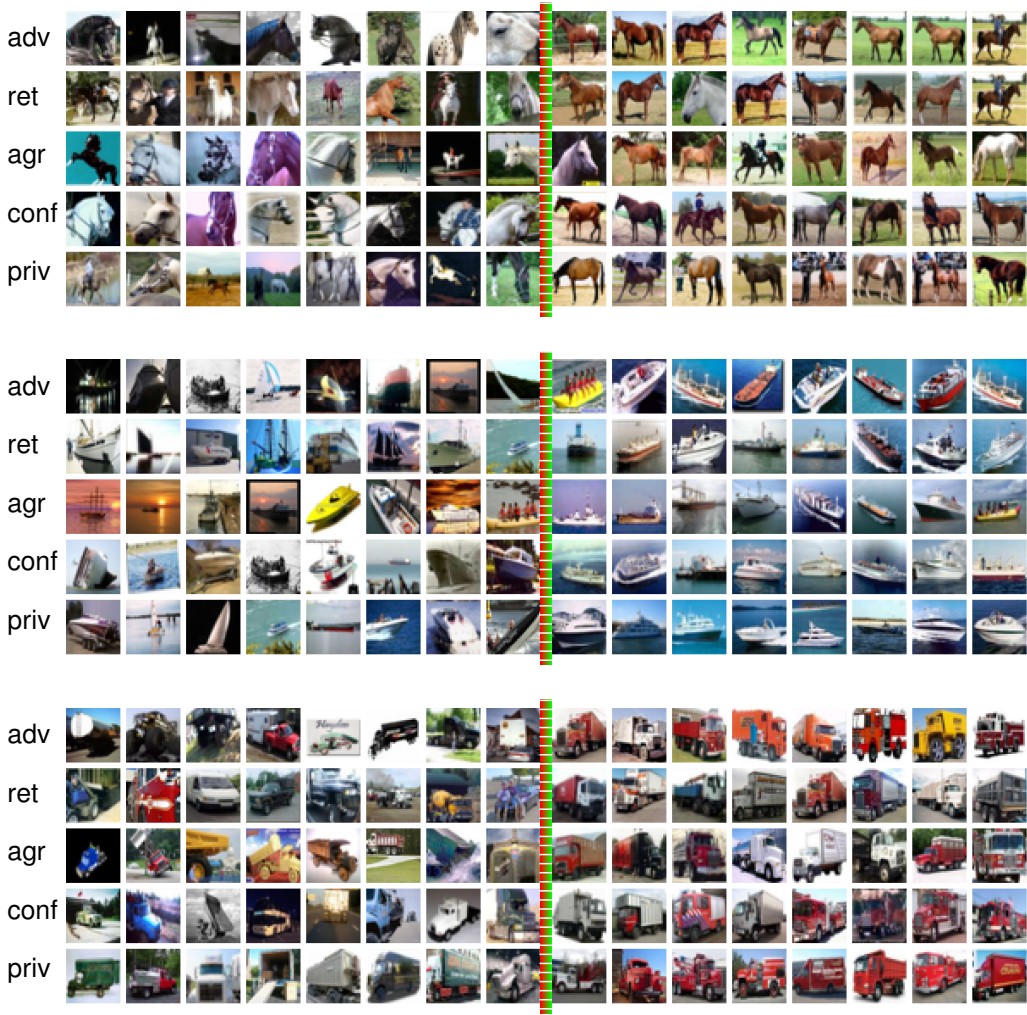

## B.4  IMAGENET EXTREME OUTLIERS AND PROTOTYPES

The following pre-trained ImageNet models were used: DenseNet121, DenseNet169, DenseNet201 InceptionV3, InceptionResNetV2, Large NASNet, Mobile NASNet, ResNet50, VGG16, VGG19, and Xception. They are all found in the Keras library: `https://keras.io/applications`.

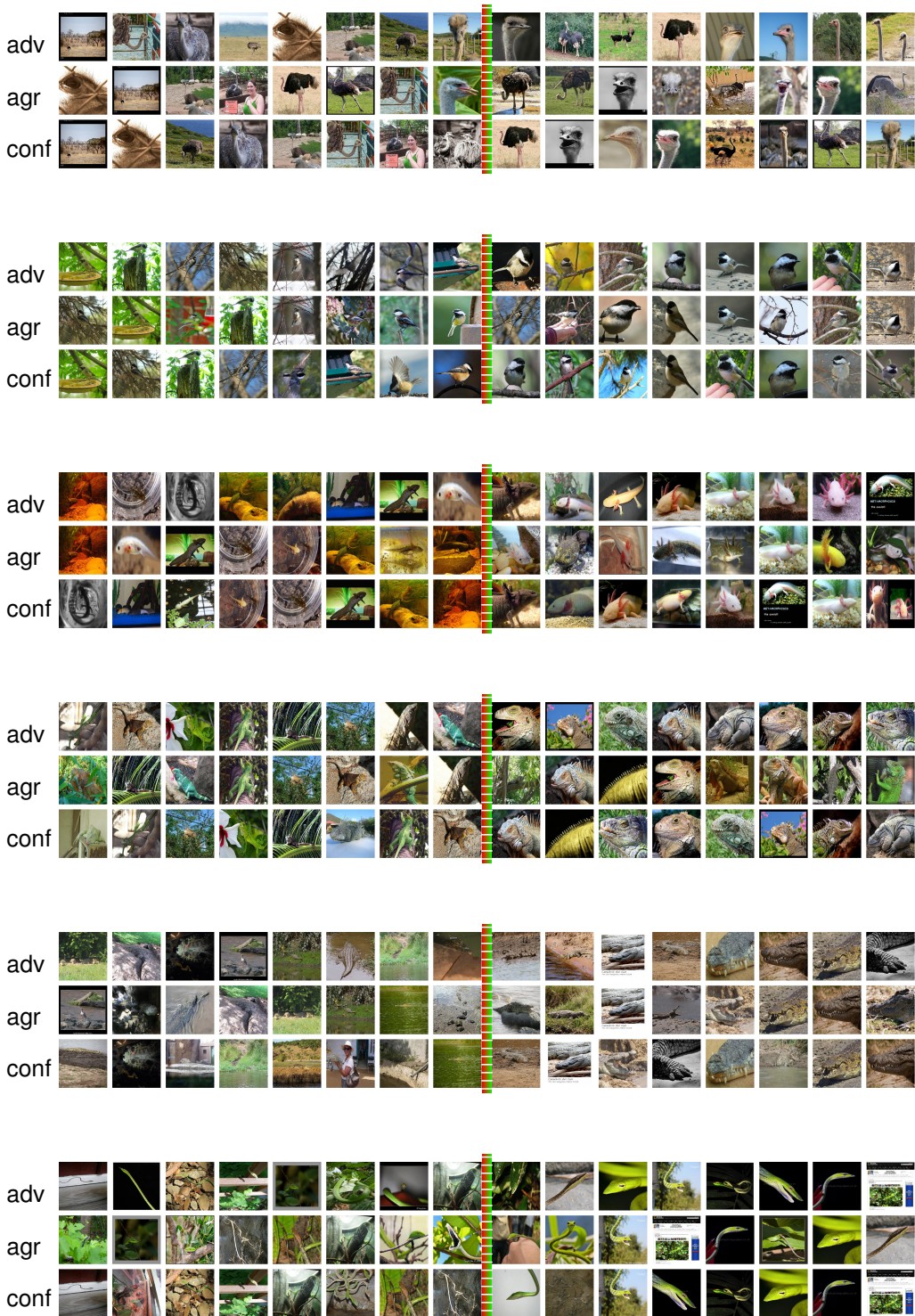

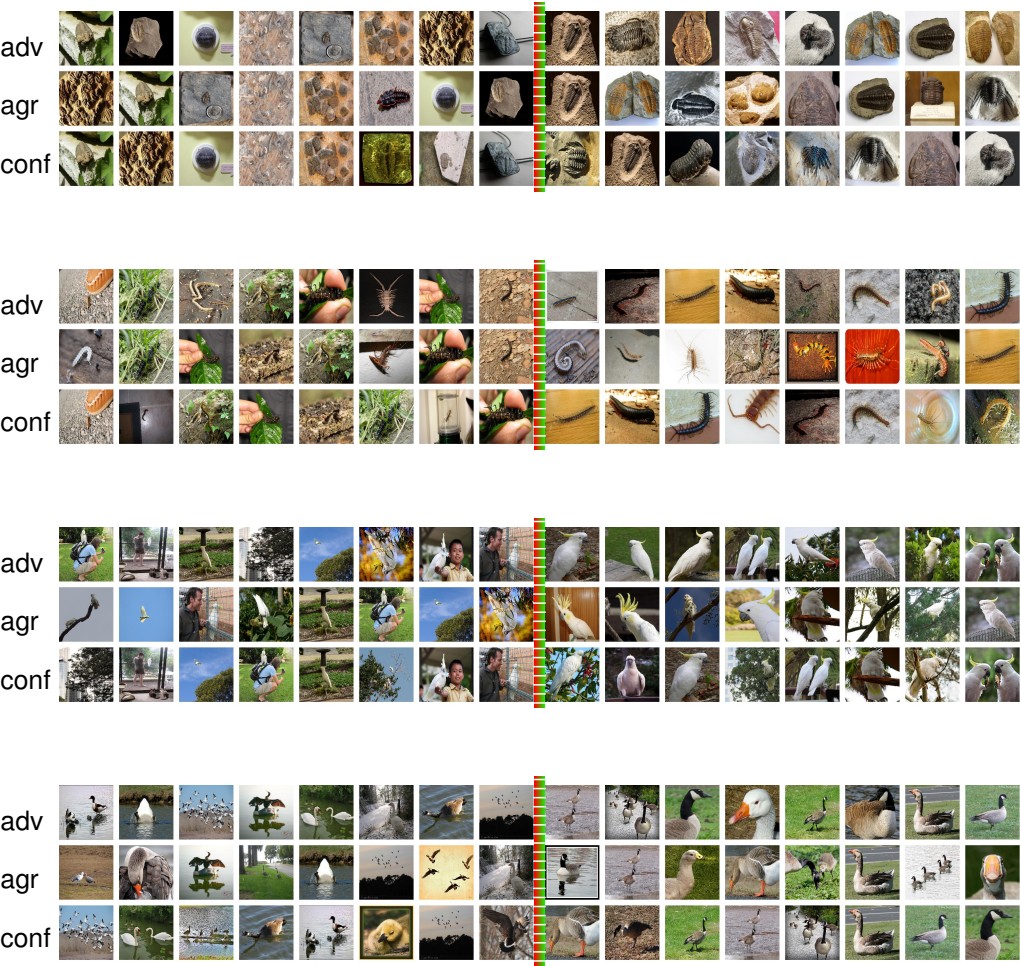

## C  PROTOTYPE ACCURACY WHEN TRAINING ON PROTOTYPES

The three matrices that follow respectively report the accuracy of MNIST, Fashion-MNIST and CIFAR-10 models learned on training examples with varying degrees of prototypicality and evaluated on test examples also with varying degrees of prototypicality. Specifically, the model used to compute cell $(i, j)$ of a matrix is learned on training data that is ranked in the $i^{th}$ percentile of adv prototypicality. The model is then evaluated on the test examples whose adv prototypicality falls under the $j^{th}$ prototypicality percentile. For all datasets, these matrices show that performing well on prototypes is possible even when the model is trained on outliers. For MNIST, this shows again that training on outliers provides better performance across the range of test data (from outliers to prototypes). For Fashion-MNIST and CIFAR-10, this best performance is achieved by training on examples that are neither prototypical nor outliers.

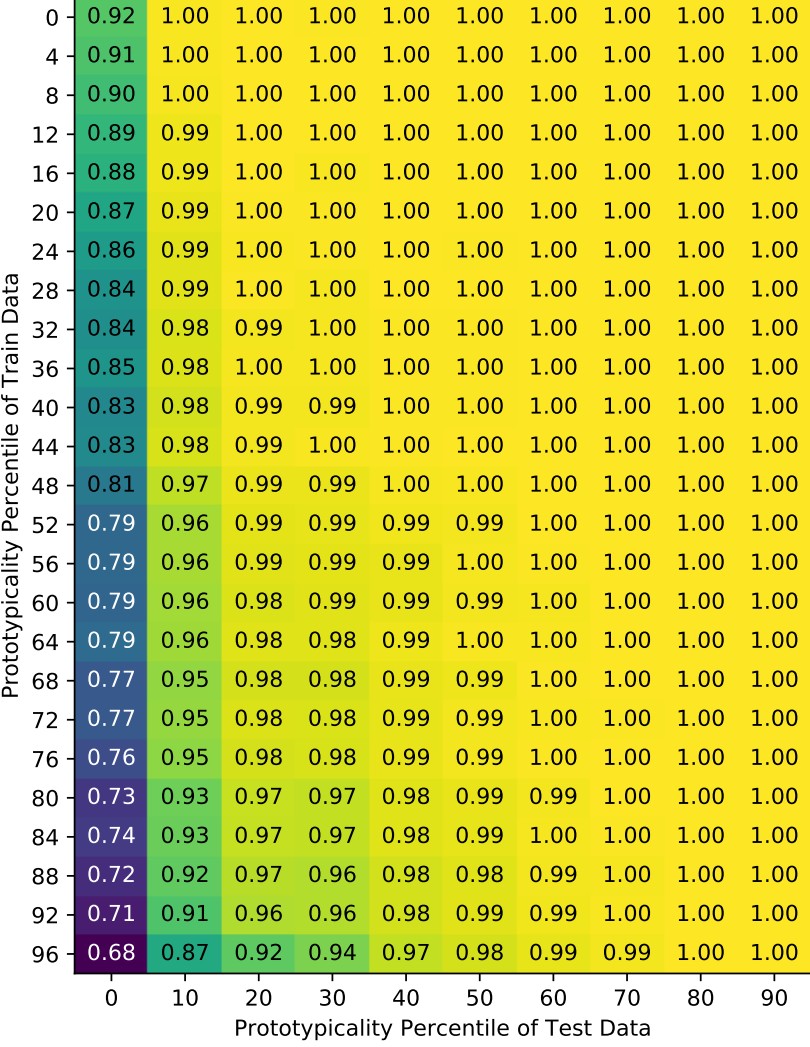

Figure 7: MNIST

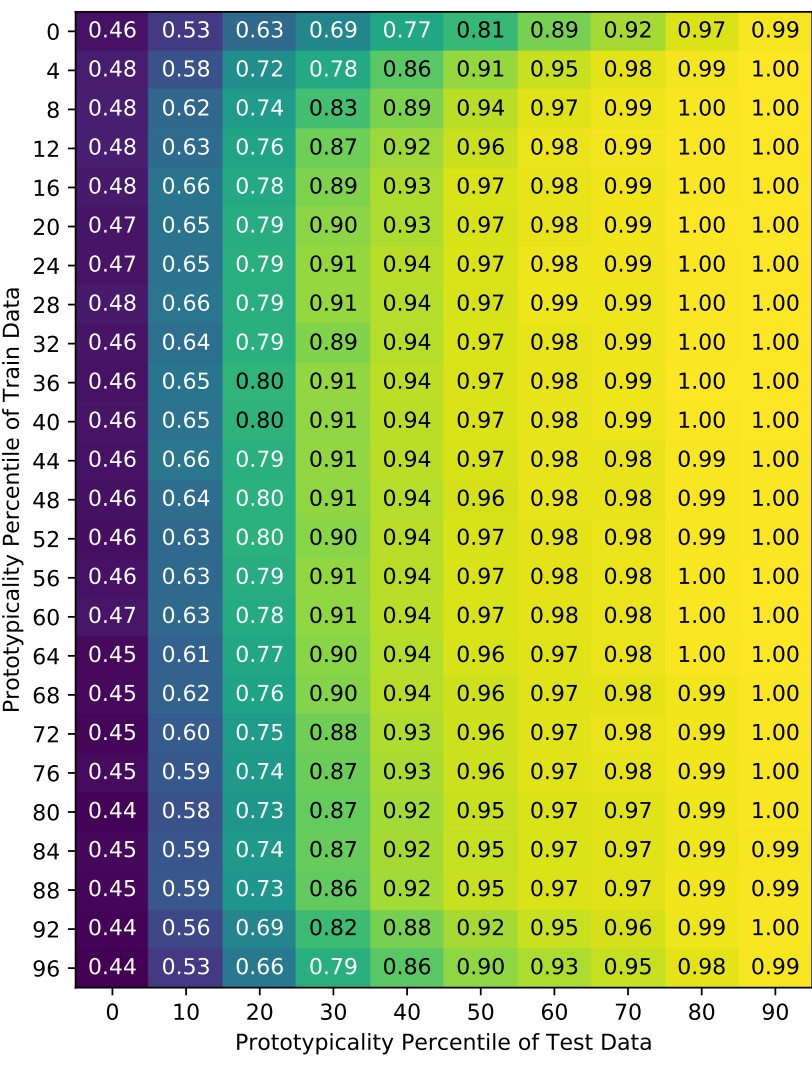

Figure 8: Fashion-MNIST

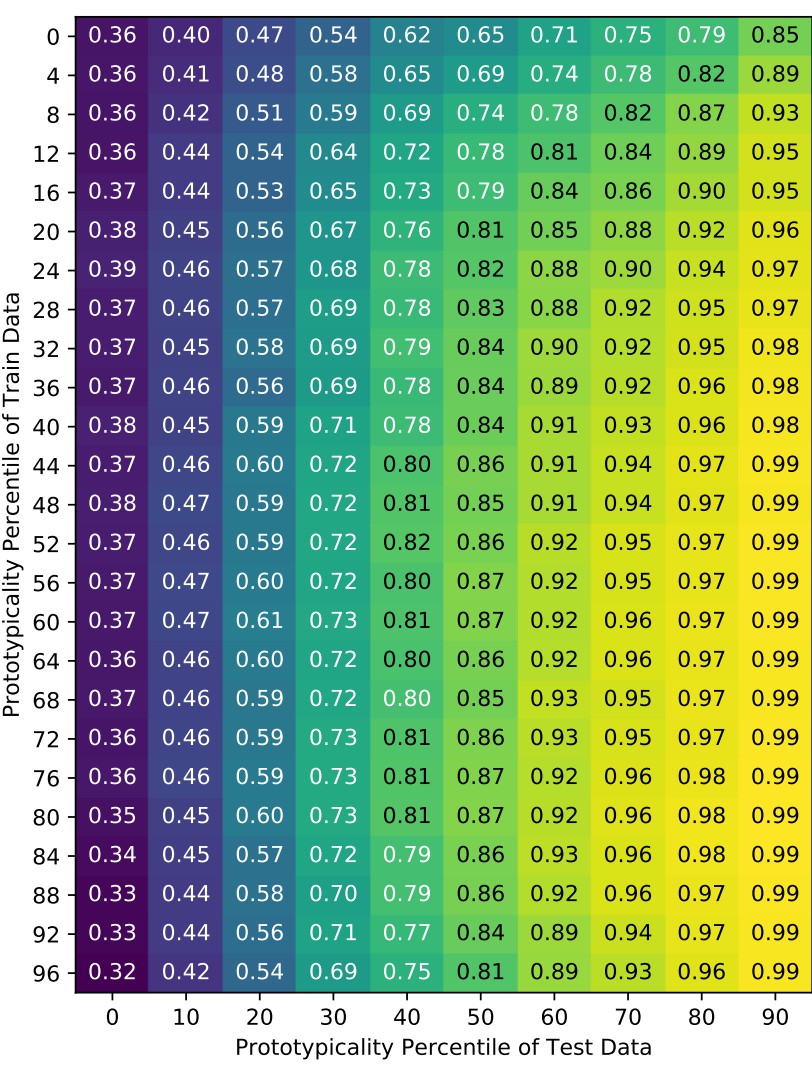

Figure 9: CIFAR-10

## D HUMAN STUDY EXAMPLE

We presented Mechanical Turk taskers with the following webpage, asking them to select the worst image of the nine in the grid.

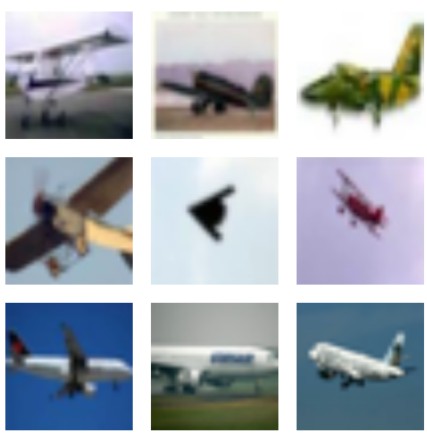

## E ADVERSARIAL ROBUSTNESS OF MODEL TRAINED ON PROTOTYPES

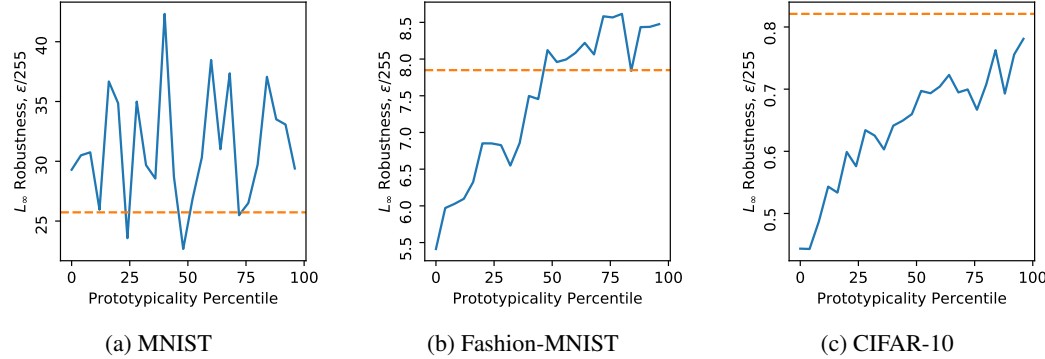

(a) MNIST          (b) Fashion-MNIST          (c) CIFAR-10

Figure 10: The blue curves indicate the training accuracy of models trained on slices of $5,000$ examples selected according to their prototypicality—as reported on the x-axis. A baseline, obtained by training the model on the entire dataset is indicated by the dotted-orange line. Models trained on prototypes on Fashion-MNIST and CIFAR-10 are $2\times$ more robust to adversarial examples, when training on slices of $5,000$ prototypical examples as opposed to slices of $5,000$ outlier examples. On MNIST there is no significant difference; almost all examples are good.

## F    REVEALING AND CLUSTERING INTERESTING EXAMPLES AND SUBMODES

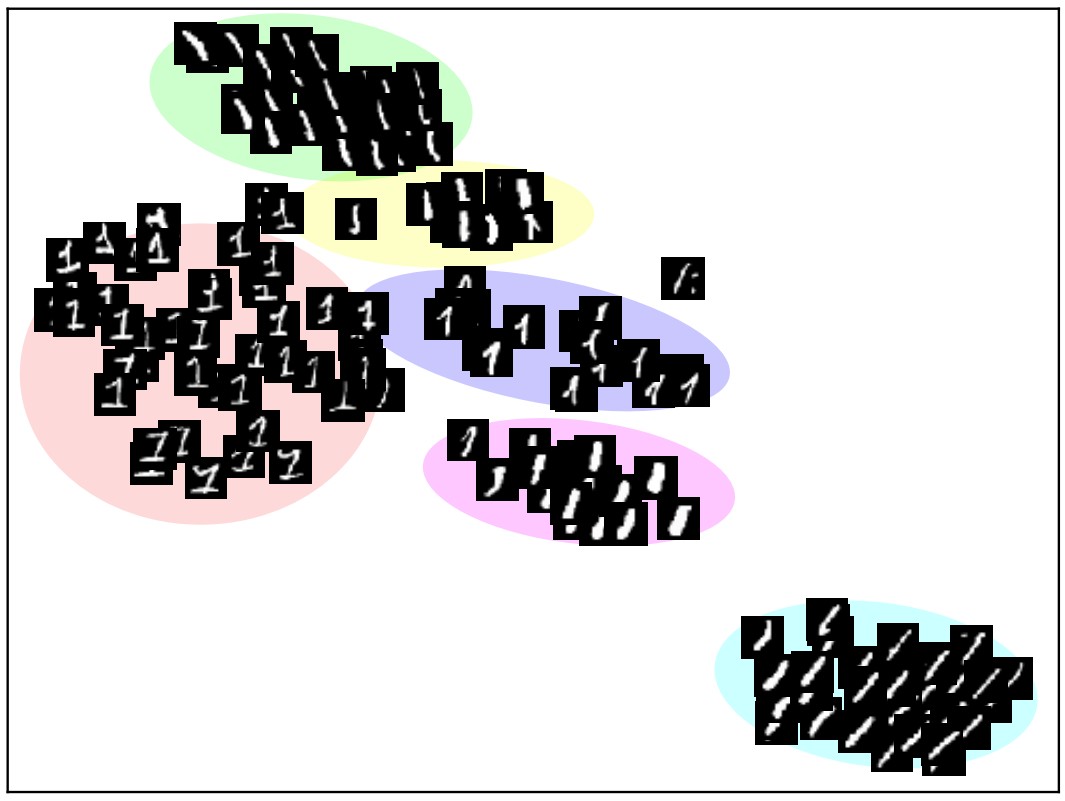

Figure 11: Uncommon submodes found within the MNIST "1" class, and their HDBSCAN clusters.

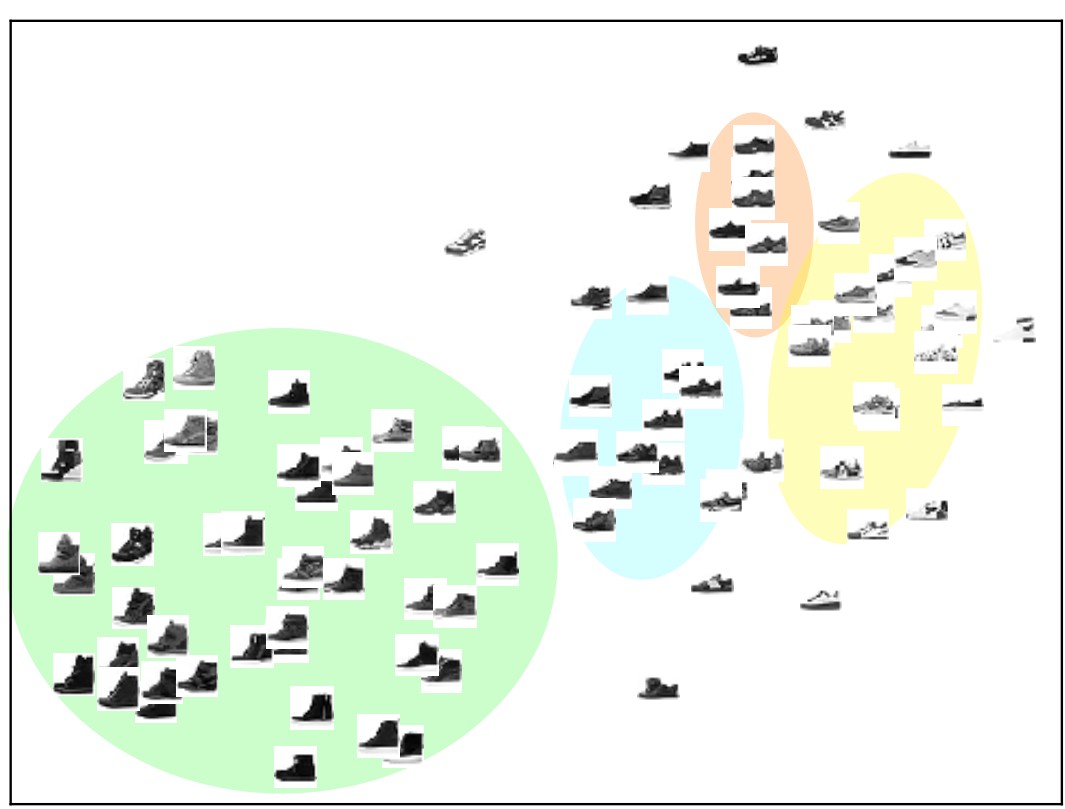

Figure 12: Memorized exceptions in the Fashion-MNIST "sneakers," and their HDBSCAN clusters.

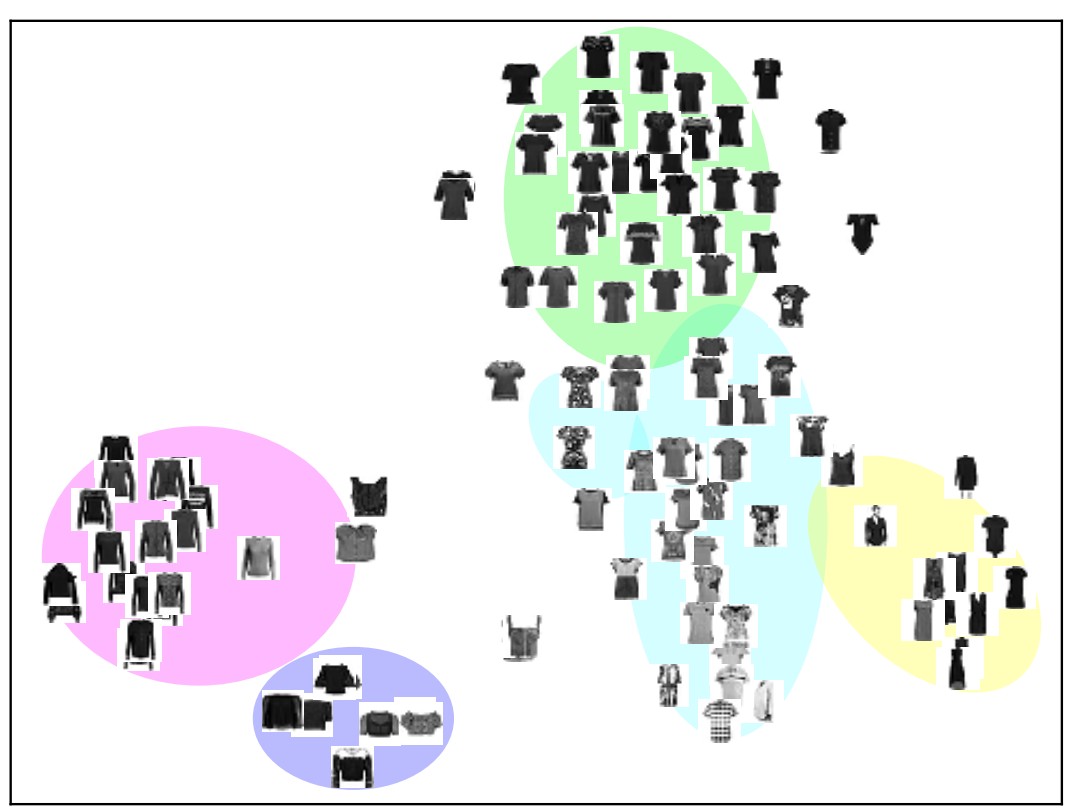

Figure 13: Memorized exceptions in the Fashion-MNIST "shirts,", and their HDBSCAN clusters.

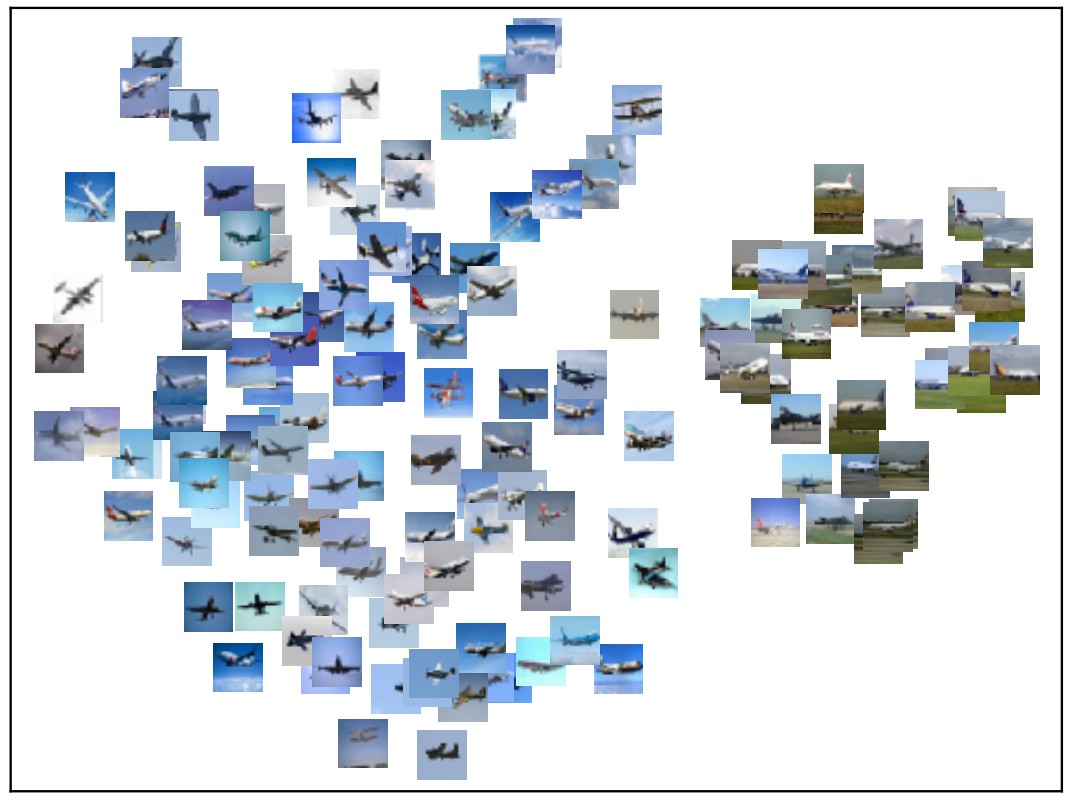

Figure 14: Canonical prototypes in the CIFAR-10 "airplane" class.

# G COMPARING DENSITY OF PROTOTYPICALITY RANKINGS FOR DIFFERENT METRICS OVER THE OUTPUT CLASSES OF ALL LEARNING TASKS

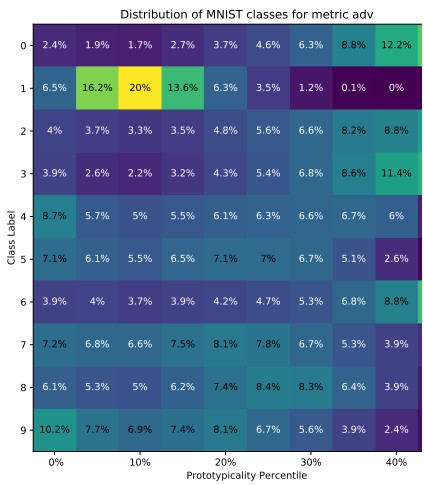

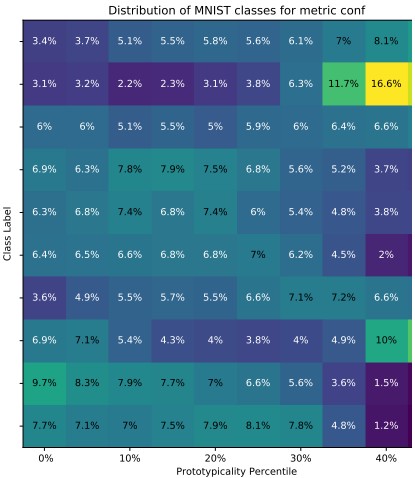

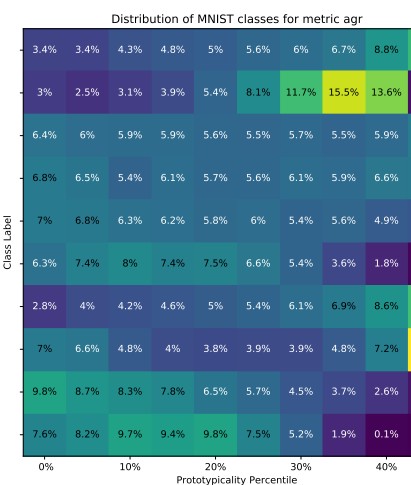

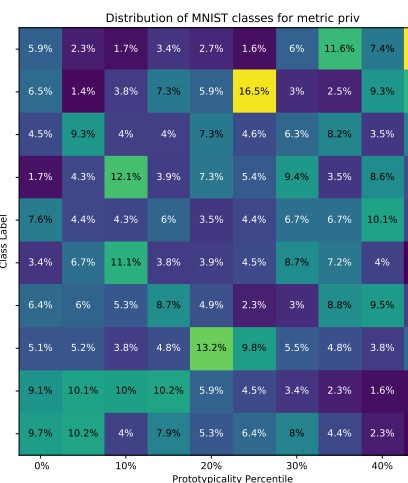

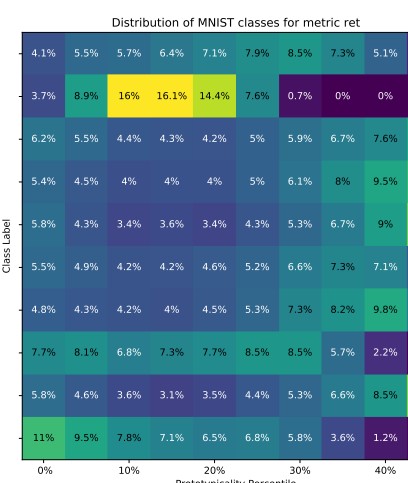

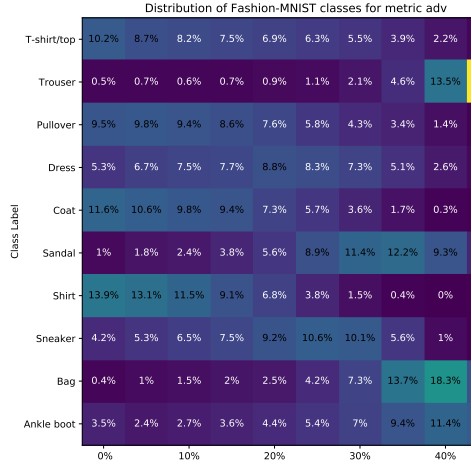

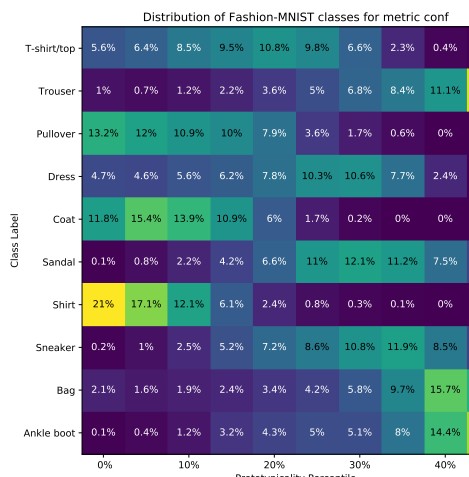

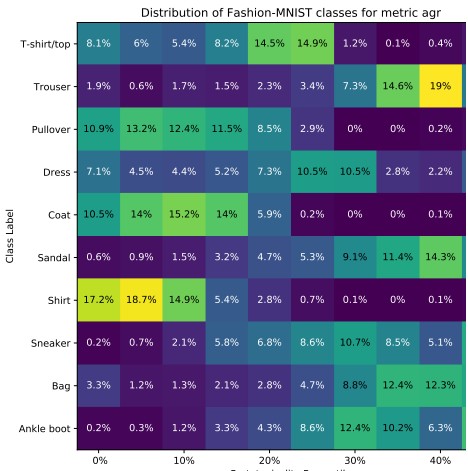

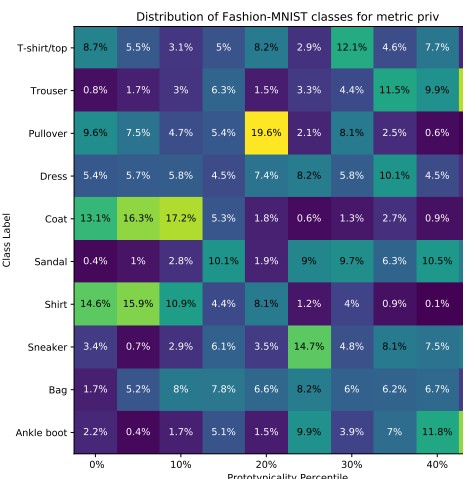

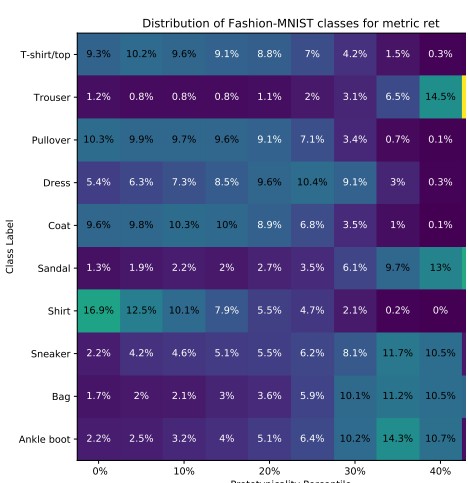

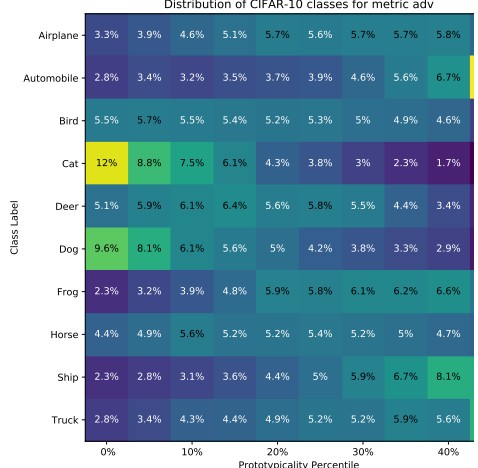

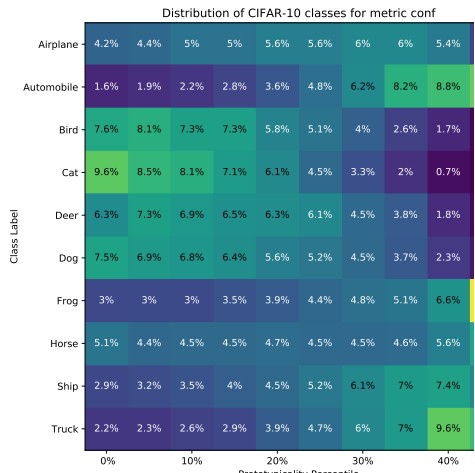

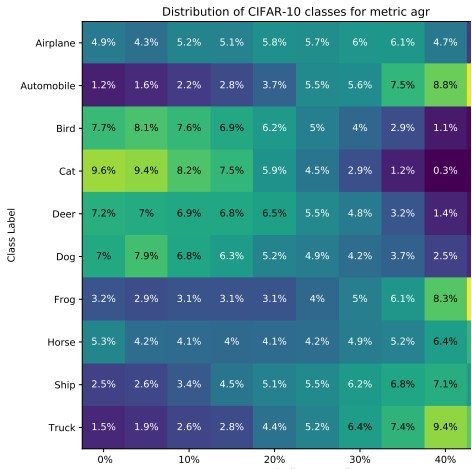

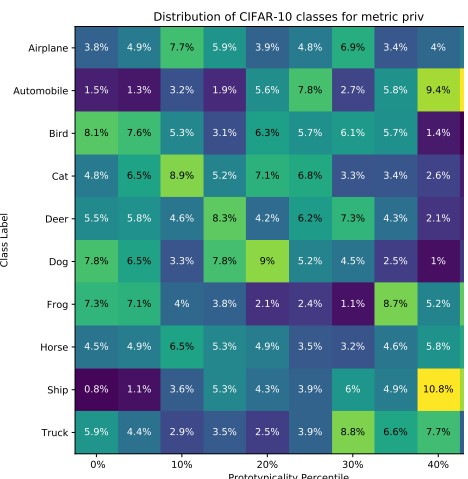

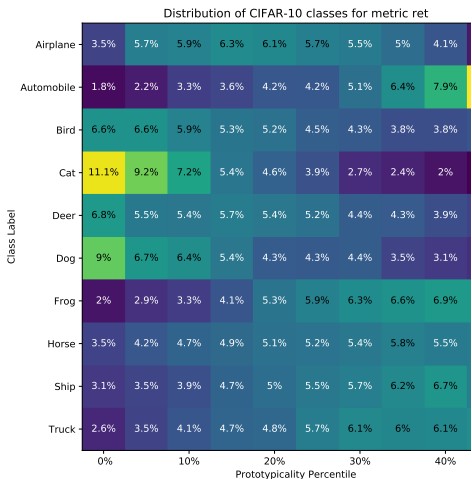

# H  ALL MEMORIZED EXCEPTIONS FOR ALL FASHION-MNIST CLASSES

Below are all the memorized exceptions, as defined in the body of the paper, for all Fashion-MNIST output classes:

- Tshirt/top
- Trouser
- Pullover
- Dress
- Coat
- Sandal
- Shirt
- Sneaker
- Bag
- Ankle boot

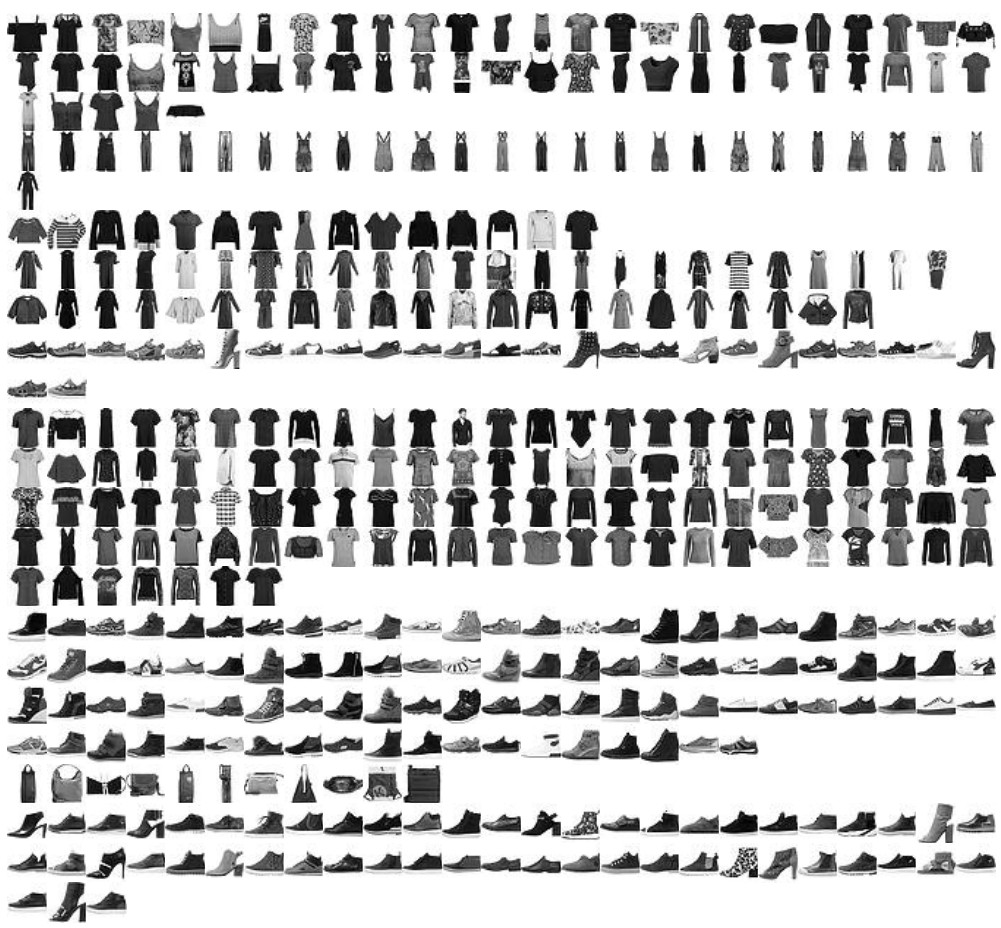

# I ImageNet Memorized Exceptions

Index 6 stingray

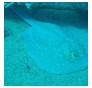

Index 7 cock

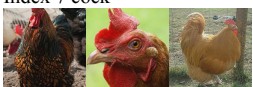

Index 8 hen

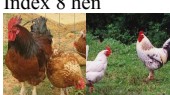

Index 32 tailed frog, bell toad, ribbed toad, tailed toad, Ascaphus trui

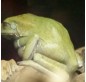

Index 36 terrapin

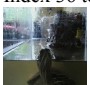

Index 40 American chameleon, anole, Anolis carolinensis

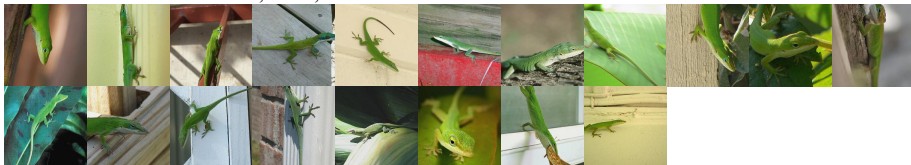

Index 46 green lizard, Lacerta viridis

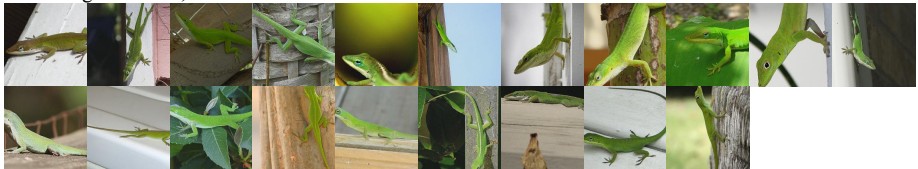

Index 66 horned viper, cerastes, sand viper, horned asp, Cerastes cornutus

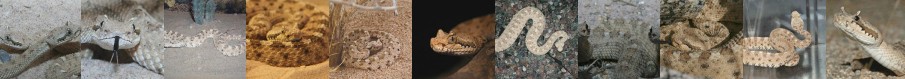

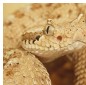

Index 68 sidewinder, horned rattlesnake, Crotalus cerastes

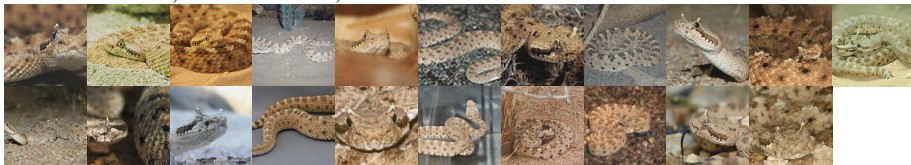

Index 72 black and gold garden spider, Argiope aurantia

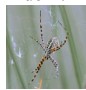

Index 101 tusker

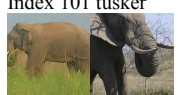

Index 103 platypus, duckbill, duckbilled platypus, duck-billed platypus, Ornithorhynchus anatinus

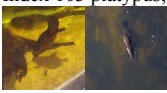

Index 122 American lobster, Northern lobster, Maine lobster, Homarus americanus

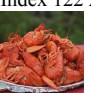

Index 124 crayfish, crawfish, crawdad, crawdaddy

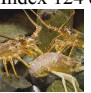

Index 126 isopod

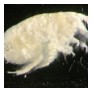

Index 161 basset, basset hound

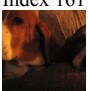

Index 166 Walker hound, Walker foxhound

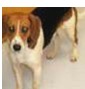

Index 167 English foxhound

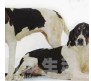

Index 170 Irish wolfhound

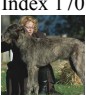

Index 179 Staffordshire bullterrier, Staffordshire bull terrier

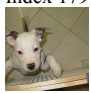

Index 180 American Staffordshire terrier, Staffordshire terrier, American pit bull terrier, pit bull terrier

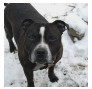

Index 196 miniature schnauzer

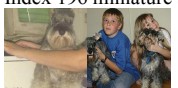

Index 197 giant schnauzer

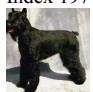

Index 198 standard schnauzer

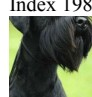

Index 206 curly-coated retriever

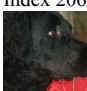

Index 214 Gordon setter

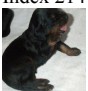

Index 223 schipperke

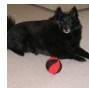

Index 238 Greater Swiss Mountain dog

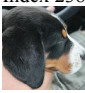

Index 248 Eskimo dog, husky

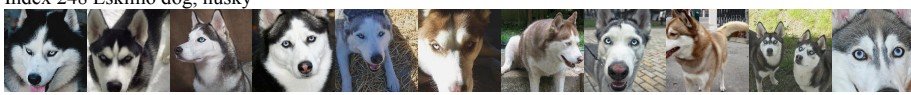

Index 250 Siberian husky

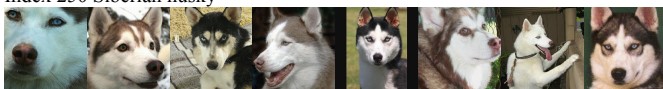

Index 264 Cardigan, Cardigan Welsh corgi

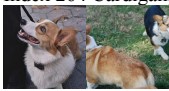

Index 265 toy poodle

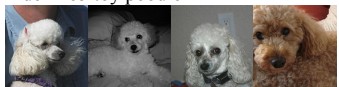

Index 266 miniature poodle

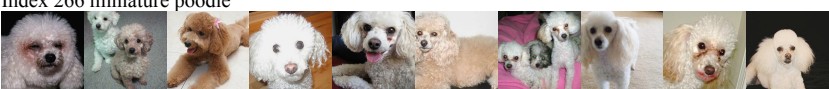

Index 270 white wolf, Arctic wolf, Canis lupus tundrarum

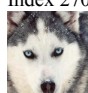

Index 278 kit fox, Vulpes macrotis

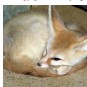

Index 290 jaguar, panther, Panthera onca, Felis onca

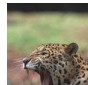

Index 304 leaf beetle, chrysomelid

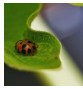

Index 311 grasshopper, hopper

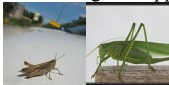

Index 312 cricket

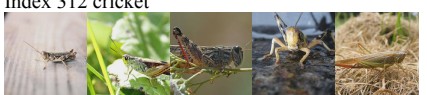

Index 319 dragonfly, darning needle, devil's darning needle, sewing needle, snake feeder, snake doctor, mosquito hawk, skeeter hawk

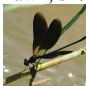

Index 320 damselfly

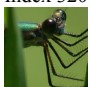

Index 334 porcupine, hedgehog

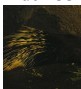

Index 341 hog, pig, grunter, squealer, Sus scrofa

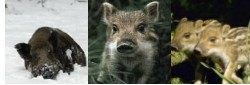

Index 342 wild boar, boar, Sus scrofa

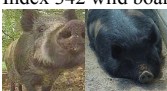

Index 345 ox

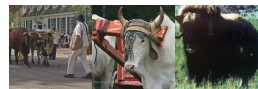

Index 348 ram, tup

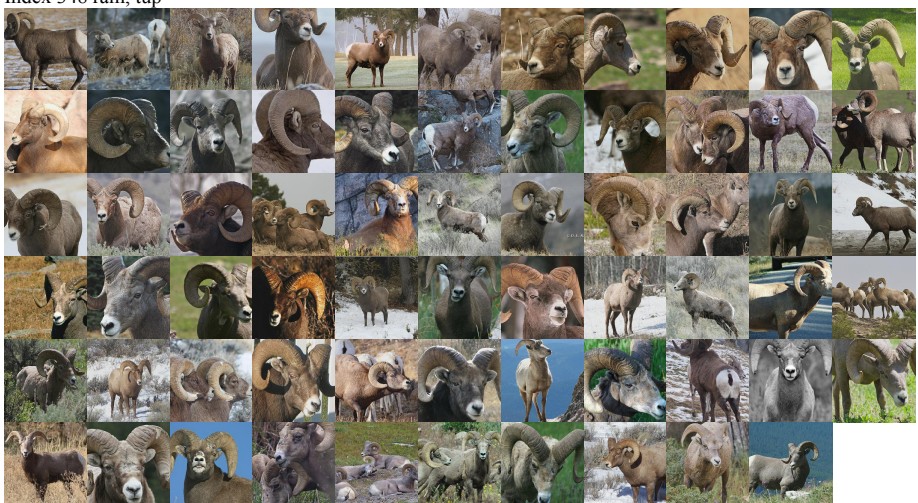

Index 349 bighorn, bighorn sheep, cimarron, Rocky Mountain bighorn, Rocky Mountain sheep, Ovis canadensis

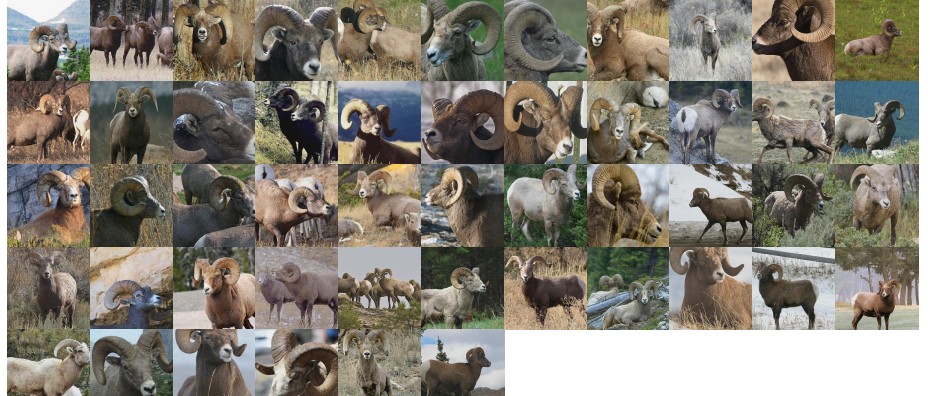

Index 359 black-footed ferret, ferret, Mustela nigripes

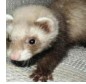

Index 380 titi, titi monkey

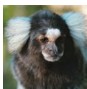

Index 383 Madagascar cat, ring-tailed lemur, Lemur catta

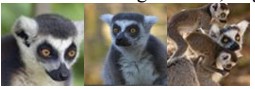

Index 386 African elephant, Loxodonta africana

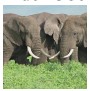

Index 390 eel

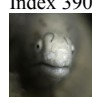

Index 399 abaya

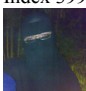

Index 400 academic gown, academic robe, judge's robe

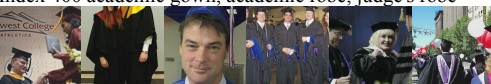

Index 409 analog clock

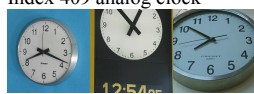

Index 413 assault rifle, assault gun

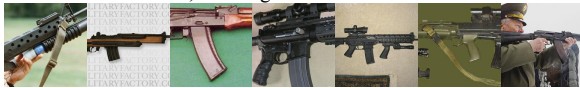

Index 417 balloon

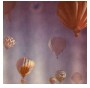

Index 419 Band Aid

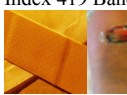

Index 423 barber chair

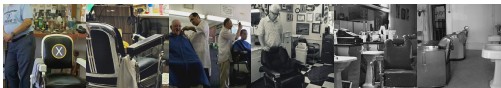

Index 424 barbershop

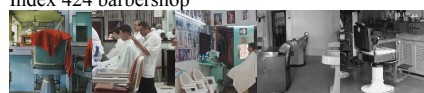

Index 429 baseball

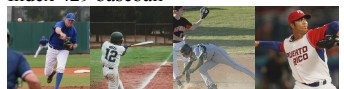

Index 434 bath towel

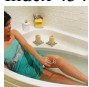

Index 435 bathtub, bathing tub, bath, tub

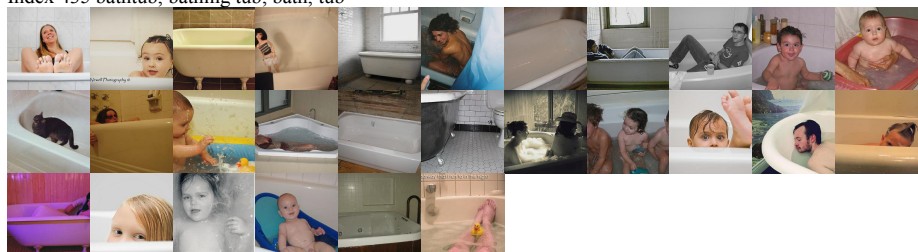

Index 440 beer bottle

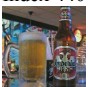

Index 461 breastplate, aegis, egis

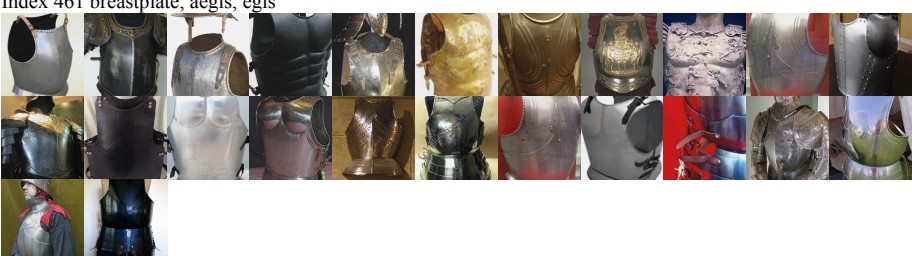

Index 465 bulletproof vest

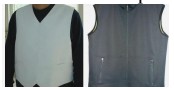

Index 479 car wheel

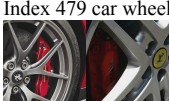

Index 484 catamaran

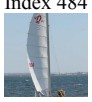

Index 505 coffeepot

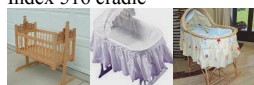

Index 516 cradle

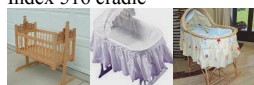

Index 524 cuirass

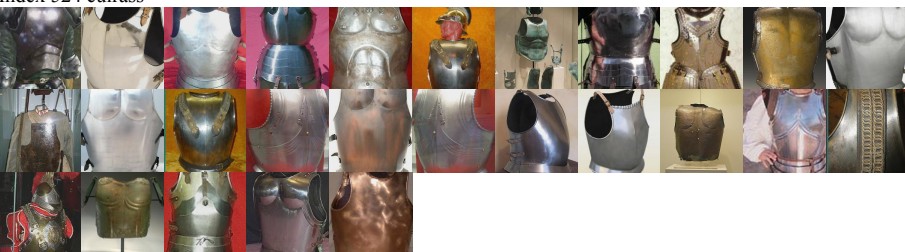

Index 538 dome

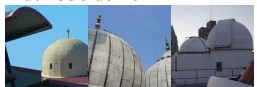

Index 541 drum, membranophone, tympan

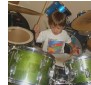

Index 550 espresso maker

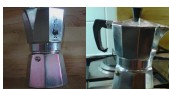

Index 579 grand piano, grand

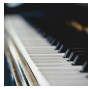

Index 583 guillotine

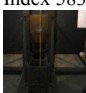

Index 591 handkerchief, hankie, hanky, hankey

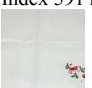

Index 595 harvester, reaper

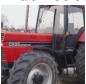

Index 604 hourglass

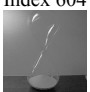

Index 619 lampshade, lamp shade

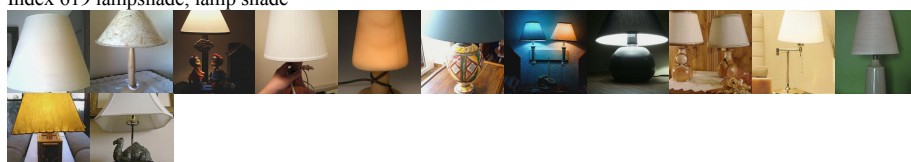

Index 620 laptop, laptop computer

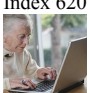

Index 636 mailbag, postbag

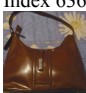

Index 638 maillot

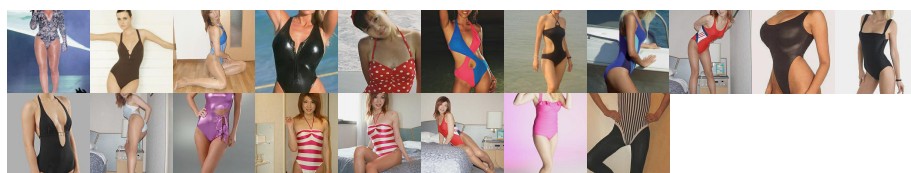

Index 639 maillot, tank suit

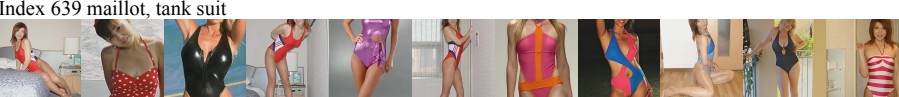

Index 643 mask

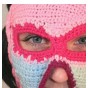

Index 647 measuring cup

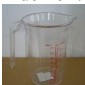

Index 657 missile

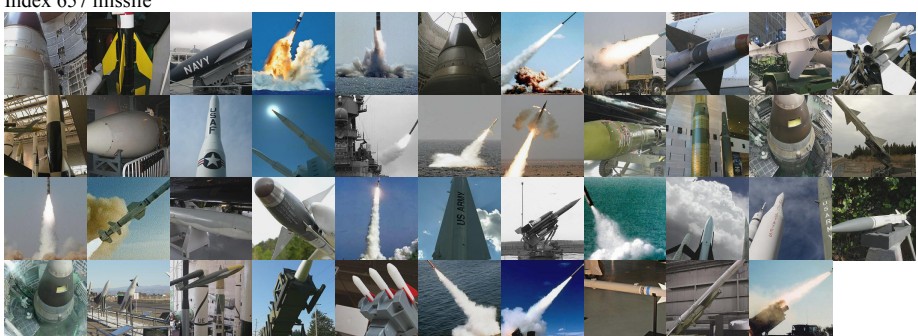

Index 665 moped

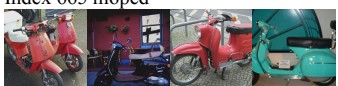

Index 667 mortarboard

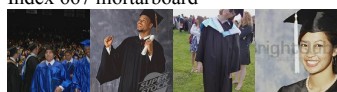

Index 668 mosque

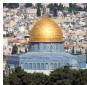

Index 670 motor scooter, scooter

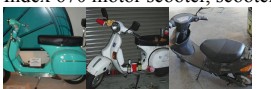

Index 678 neck brace

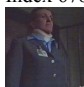

Index 681 notebook, notebook computer

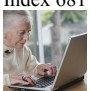

Index 700 paper towel

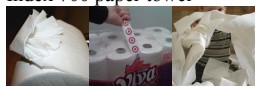

Index 739 potter's wheel

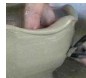

Index 744 projectile, missile

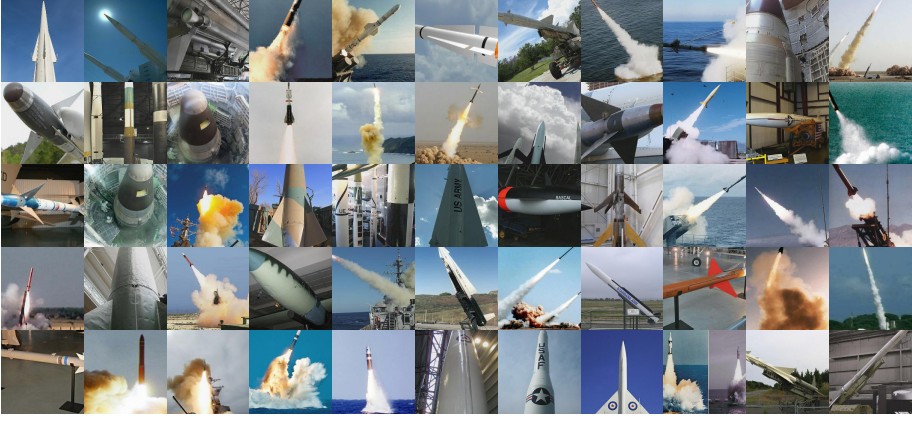

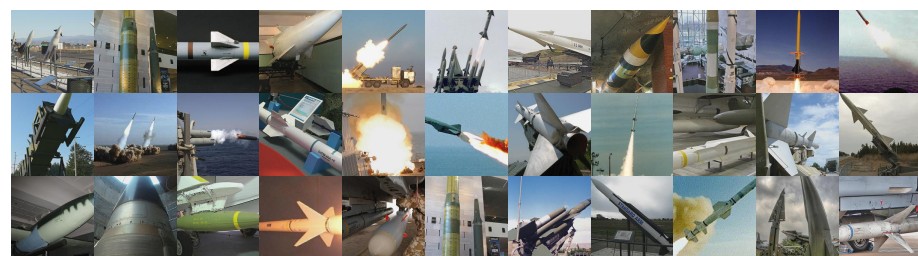

Index 748 purse

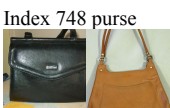

Index 764 rifle

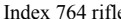
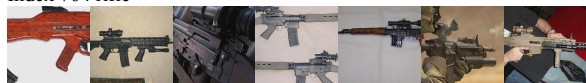

Index 804 soap dispenser

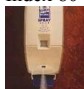

Index 808 sombrero

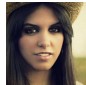

Index 810 space bar

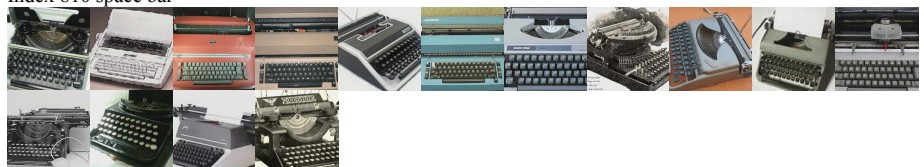

Index 817 sports car, sport car

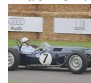

Index 827 stove

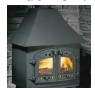

Index 830 stretcher

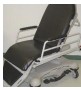

Index 836 sunglass

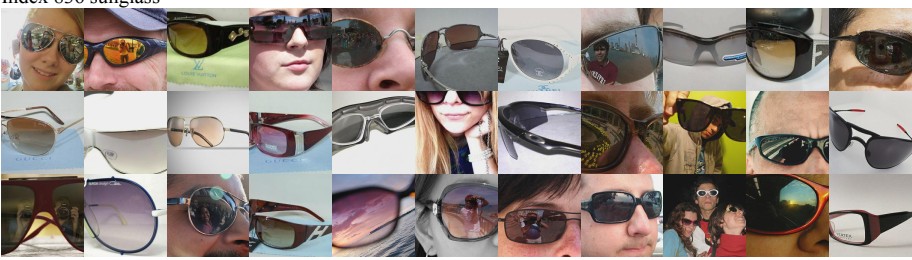

Index 837 sunglasses, dark glasses, shades

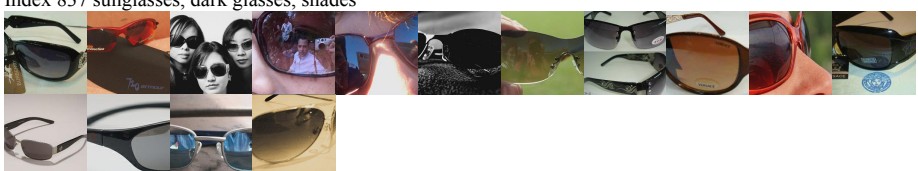

Index 841 sweatshirt

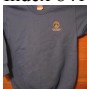

Index 842 swimming trunks, bathing trunks

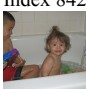

Index 846 table lamp

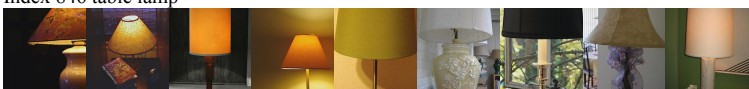

Index 847 tank, army tank, armored combat vehicle, armoured combat vehicle

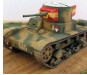

Index 876 tub, vat

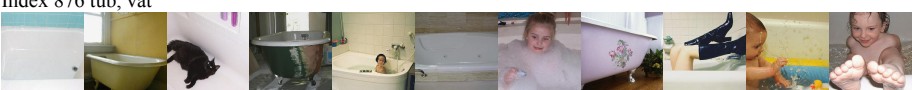

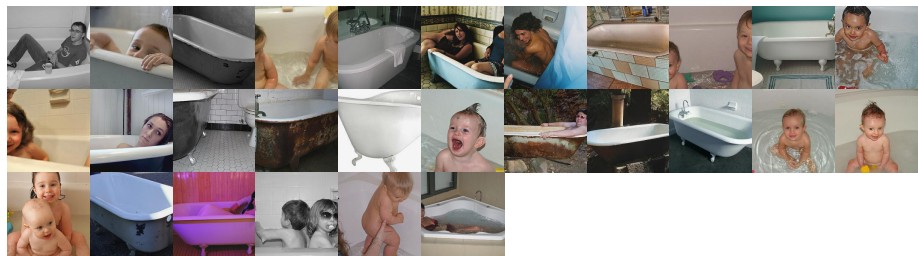

Index 878 typewriter keyboard

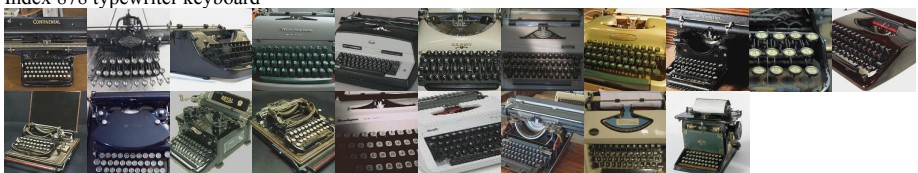

Index 892 wall clock

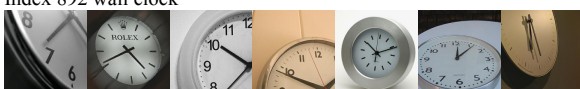

Index 903 wig

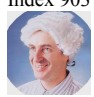

Index 907 wine bottle

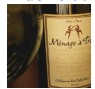

Index 914 yawl

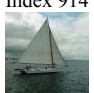

Index 925 consomme

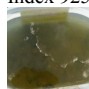

Index 928 ice cream, icecream

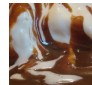

Index 939 zucchini, courgette

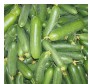

Index 954 banana

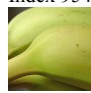

Index 960 chocolate sauce, chocolate syrup

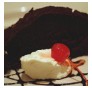

Index 961 dough

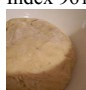

Index 962 meat loaf, meatloaf

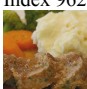

Index 966 red wine

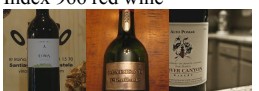

Index 981 ballplayer, baseball player

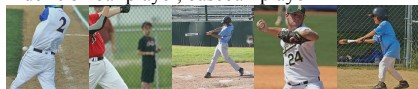

Index 987 corn

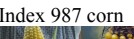
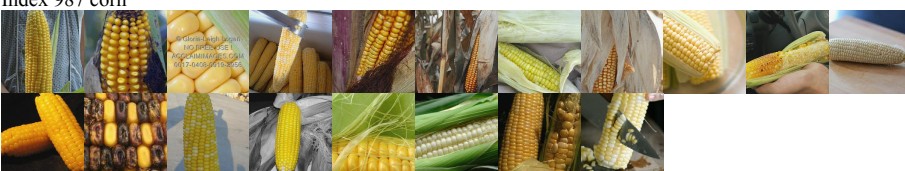

