# OpenReview forum: "Prototypical Examples in Deep Learning: Metrics, Characteristics, and Utility"
_ICLR.cc/2019/Conference_

### Official Review · AnonReviewer3 · 2018-11-02
**Prototypical Examples on Small Datasets**

**Rating:** 5
**Confidence:** 4

**Review:**

## Strength

This paper explores ways of identifying prototypes with extensive qualitative and quantitative empirical attempts.

## Weakness

### Not practical

The authors report that “removing individual training examples did not have a measurable impact on model performance”. However, this seems not to be supported by experiments.
First, it is not clear what exactly models do they use in Section 4, e.g. ResnetV2 with how many layers? Learning rate schedules?
Second, why is the baseline models on CIFAR-10 perform so bad (<90%) even with 100% data?
Third, with `"adv" metric, we need to perform adversarial-example attacks before training, which has little value in practice.

### Datasets

They only conduct quantitative experiments (section 4) on relatively small datasets (i.e. MNIST, Fashion-MNIST and CIFAR-10). It is not clear how it will generalize to more realistic settings.

## Most confusing typos

1. Section 4, paragraph 5, "However, we find that training only on the most prototypical examples gives extremely high accuracy on the other prototypical examples." Is there a missing "than"? It's confused.
2. The description of Figure 6 is not clear enough. Especially there is no explanation to (d, e, f).

---

> ### Author Response · Authors · 2018-11-07
> **Regarding the comment "not practical"**
>
> We are unsure whether the reviewer's concerns about practicality are with respect to our methods or our results.
>
> Our methods are practical and are simple to implement with less than ten lines of code (either by querying the pre-trained model, making calls to CleverHans for generating adversarial examples, or reusing the existing training code for retraining). The exception to this is our prototypicality metric based on privacy, which we agree is a state-of-the-art technique. We will release our code for this once we have completed the experiments.
>
> Our results have several practical benefits:
> 1. We provide a practical technique for identifying mislabeled training examples (see for example Appendix H) which can help automate the collection of training data.
> 2. We provide a practical technique for identifying inherently ambiguous training examples such as the boot-like sneakers in the Fashion-MNIST dataset (see Figure 12 in Appendix F), which can be used in training corpus data constructions and class definitions. For example in Appendix J that we have added in response, we show that our techniques for finding memorized exceptions apply to ImageNet. Our methods can automatically find many classes that are inherently ambiguous or are overlapping on ImageNet (“tusker” vs. “elephant”; “sunglass” vs. “sunglasses, dark glasses, shades”; “projectile, missile” vs. “missile”; “maillot” vs. “maillot, tank suit”; “breastplate, aegis, egis” vs. “cuirass”).
> 3. We provide a practical technique for identifying uncommon sub-modes (e.g., 1s written in both serif and non-serif style), see Figure 11 Appendix F, which can be used to understand and balance a training data set.
> 4. We demonstrate four new practical metrics for identifying prototypical examples that correspond to human intuition (as well as confirming the prior work of Stock and Cisse (2018)), See Appendix G.
> 5. We find that training on prototypical examples is a quick and efficient method (requiring only 2%-10% of the original training data) for constructing models that perform well on test prototypical examples.

---

> > ### Comment · AnonReviewer3 · 2018-12-02
> > **The cost of the trained model for generating adversarial examples**
> >
> > Being not practical refers to Section 4, utilizing prototypes to improve aspects of machine learning. It is true that generating adversarial examples is relatively efficient when given access to a trained model, but, if I'm not wrong, this model has to be trained on all the examples. It seems that the author did not take the training of this model into account, which is why I consider it  not practical.

---

> ### Author Response · Authors · 2018-11-07
> **Response to review**
>
> Thank you for your review. Below we respond inline to each point you raise.
>
> > Prototypical Examples on Small Datasets
>
> Our experimental results include ImageNet, which is usually not considered a small dataset. As discussed below, have already added more ImageNet results to the revised paper and will provide complete results in an upcoming revision.
>
>
> > The authors report that “removing individual training examples did not have a measurable impact on model performance”. However, this seems not to be supported by experiments.
>
> We clarified this statement in the related work section to say the following: “Conversely, for MNIST, we found in our experiments that removing individual training examples did not have a measurable impact on the predictions of individual test examples. Specifically, we trained many models to 100% training accuracy where we left one training example out for each model. There was no statistically significant difference between the models predictions on each individual test example.”
>
>
> > First, it is not clear what exactly models do they use in Section 4, e.g. ResnetV2 with how many layers? Learning rate schedules?
>
> We trained a ResNet-20 for 100 epochs, all other hyperparameters are unchanged from
> https://raw.githubusercontent.com/keras-team/keras/master/examples/cifar10_resnet.py
> We have included these details in the revised Appendix B.3.
>
>
> > Second, why is the baseline models on CIFAR-10 perform so bad (<90%) even with 100% data?
>
> Because we had to train so many models (~100) we trained for fewer epochs each. This made some models reach lower than 90% accuracy. We will resume training for these models to reach higher accuracy.
>
>
> > Third, with `adv` metric, we need to perform adversarial-example attacks before training, which has little value in practice.
>
> We are a bit confused what the reviewer may mean here. To clarify: we do not need to perform any adversarial training, nor are we trying to find the robustness of a given model. Concretely, we do not change anything that happens either before or during training based on adversarial techniques. Instead, only after the model is trained do we follow Stock and Cisse (2018) and measure the distance from some input to the decision boundary by finding the smallest adversarial perturbation.
>
>
> ### Datasets
>
> > They only conduct quantitative experiments (section 4) on relatively small datasets (i.e. MNIST, Fashion-MNIST and CIFAR-10). It is not clear how it will generalize to more realistic settings.
>
> In the paper we show quantitative results on ImageNet in Figure 1(d), and qualitative results in Appendix B.4. To stress that our results hold for ImageNet, we have also added all of the memorized exceptions for ImageNet in Appendix J. In the next revision of our paper we will add the remaining metrics (ret and priv). Both qualitatively and quantitatively our results on ImageNet are matching those for CIFAR-10 and FashionMNIST in all of our experiments so far. We are currently finalizing the differentially private training results on ImageNet models. This has required us to overcome several challenges (not just the computational constraints of training ~10 ImageNet models). Finding the right set of hyperparameters to successfully train a differentially private ImageNet model is a novel contribution: no prior work has ever trained an ImageNet model with differential privacy.
>
> We are also running curves similar to Figure 6 on ImageNet, but since each curve requires training 20 ImageNet models this is taking some time.
>
>
> ## Most confusing typos
>
> > 1. Section 4, paragraph 5, "However, we find that training only on the most prototypical examples gives extremely high accuracy on the other prototypical examples." Is there a missing "than"? It's confused.
>
> We have re-phrased this sentence to be more specific. It now reads as “However, we find that training only on the most prototypical examples found in the training data gives extremely high test accuracy on the prototypical examples found in the test data."
>
>
> > 2. The description of Figure 6 is not clear enough. Especially there is no explanation to (d, e, f).
>
> We agree and have clarified and added content in this caption in red text to highlight what is new.

---

> > ### Comment · AnonReviewer3 · 2018-11-15
> > **ImageNet analysis**
> >
> > I was aware of the fact that this paper has included ImageNet dataset. By "small datasets", I meant in Figure 2, 3, 4, 5, 6, there was no analysis on ImageNet. It would be good to include more analysis on ImageNet.

---

> > ### Comment · AnonReviewer3 · 2018-12-02
> > **Section 4**
> >
> > It may seem not "creative" to ask for ImageNet results. However, this paper is positioned as an empirical study paper and CIFAR-10 only has 50k training samples with a SOTA test accuracy of 99% [GPipe: Efficient Training of Giant Neural Networks using Pipeline Parallelism](https://arxiv.org/abs/1811.06965). In other words, the datsets used in Section 4 are not very representative and cannot provide enough insights to the community. They did include the qualitative results for "priv" method. But for the experiments shown in Figure 6 (i.e. quantitative results for "adv"), in the latest version I still do not see ImageNet results.
> >
> > Actually in [Distilling the Knowledge in a Neural Network](https://arxiv.org/abs/1503.02531), the authors show that with only 3% of the data and soft labels (generated by a trained model), a student model can achieve similar performance -- they show this using 20M examples. I'm not saying the authors have to train an extreme huge model on ImageNet, but at least they should try a relatively small ResNet on ImageNet (or another decent dataset) for "adv" method.

---

### Official Review · AnonReviewer2 · 2018-11-12
**Interesting attempt at understanding prototypes but needs more work**

**Rating:** 5
**Confidence:** 4

**Review:**

Summary: This paper attempts to better understand the notion of prototypes and in some sense create a taxonomy for characterizing various prototypicality metrics. While the idea of thinking about such a taxonomy is novel, I think the paper falls in clearly justifying certain design choices such as why are the properties outlined at the beginning of Section 2 desirable. I also felt that the paper is resorting to rather informal ways of describing various properties and metrics without precisely quantifying them.

Pros:
1. Novel attempt at understanding prototypes. Two specific contributions: a) outlining the properties desirable in prototypicality metrics b) proposing new prototypicality metrics and demonstrating the relevance of the various prototypicality metrics.
2. Detailed experimental analysis along with some user studies

Cons:
1. An important drawback of this paper is that the notion of prototype is not very clearly contextualized and explained. There is often a purpose associated with identifying prototypes - are we summarizing a dataset? are we thinking about helping humans understand the behavior of a specific learning model? Answers to these questions guide the process of choosing prototypes. However, this paper seems to approach the problem of choosing prototypes via the "one approach fits all" strategy which I am not sure is even possible.
2. The choice of desirable properties is not clearly justified (Beginning of Section 2). For instance, why should prototypes be independent of learning tasks?
3. Lack of rigor in defining prototypicality metrics as well as properties in Section 2. For example, wouldn't it be possible to theoretically prove that the metrics outlined in Section 2 satisfy the desired properties?

Detailed Comments:
1. I would strongly encourage the authors to illustrate using examples in the introduction the significance of finding prototypes. What are the end goals for which these prototypes would be used? Why do you think the metric for chooosing prototypes should be independent of the learning task or model?
2. Along the same lines as the comment above, please provide detailed justifications for the list of properties provided in the beginning of Section 2. It would be even better if you could formalize these a bit more.
3. Would it be possible to theoretically show that the metrics defined in Section 2 satisfy any of the desirable properties highlighted in Section 2?

---

> ### Author Response · Authors · 2018-11-14
> **Response to Review (pt. 2)**
>
>
> > Cons:
> > 1. An important drawback of this paper is that the notion of prototype is not very clearly contextualized and explained. There is often a purpose associated with identifying prototypes - are we summarizing a dataset? are we thinking about helping humans understand the behavior of a specific learning model? Answers to these questions guide the process of choosing prototypes. However, this paper seems to approach the problem of choosing prototypes via the "one approach fits all" strategy which I am not sure is even possible.
>
> We tried to offer more clear definitions than what we found in earlier work. Unlike that work, we did not start off with a specific goal; rather, we wanted to see if five new-and-old techniques for ranking training and test examples would give insights into ML model training processes and data corpora. We explicitly did not believe that “one approach fits all,” and hence we evaluated five different techniques (actually more, but others were less informative). This was fortunate, because the differences between the metrics actually proved to be more informative than the metrics themselves, e.g., in finding memorized exceptions or mislabeled and inherently-ambiguous training data.
>
> > 2. The choice of desirable properties is not clearly justified (Beginning of Section 2). For instance, why should prototypes be independent of learning tasks?
>
> See our comment above on that list.  On this specific property, it seemed preferable to us if a single measure (i.e. mechanism for evaluating a metric) could be applied equally to classification models, generative sequences models, etc. That way, a single technique could be used to find prototypical examples in a range of different modes and for many different types of tasks. This is not a property that holds for all of our five metrics, e.g., since a notion of “confidence” or “adversarial class” simply isn’t defined in all learning tasks (e.g., training an embedding). But for both our retraining-distance and privacy-based metrics, it should be possible to apply the metric to nearly all learning tasks. Again, this seemed preferable.
>
> > 3. Lack of rigor in defining prototypicality metrics as well as properties in Section 2. For example, wouldn't it be possible to theoretically prove that the metrics outlined in Section 2 satisfy the desired properties?
>
> As said above, we tried to give a more precise definition for “prototype” than what we could find in the existing literature. We agree that our definitions are still less rigorous than would be ideal. We do not think it would be feasible to theoretically prove that our metrics satisfy our properties, since they quantitatively depend on data corpora and ML models and tasks, and some such combinations (e.g., artificially constructed ones) may surely fail the properties. However, for some existing concrete data corpus and ML model/taks, like MNIST, CIFAR-10, ImageNet, etc., we can try to empirically validate how the properties apply to our metrics, for example as we do with our human studies.  In the final revision of the paper, we will add an appendix showing how each property is supported for each of our metric.
>
> > Detailed Comments:
> > 1. I would strongly encourage the authors to illustrate using examples in the introduction the significance of finding prototypes. What are the end goals for which these prototypes would be used?
>
> In addition to performance benefits, via curriculum learning etc., discussed in Section 4, our end goals are to understand aspects of training data and tasks such as those shown in Figure 5.  In particular, we know of no other technique for finding memorized exceptions or mislabeled and inherently-ambiguous training data that works in the same way and equally well.  We will move that figure earlier in the paper, into the introduction.
>
> > Why do you think the metric for chooosing prototypes should be independent of the learning task or model?
>
> See detailed answer above.
>
> > 2. Along the same lines as the comment above, please provide detailed justifications for the list of properties provided in the beginning of Section 2. It would be even better if you could formalize these a bit more.
>
> See answer above.
>
> > 3. Would it be possible to theoretically show that the metrics defined in Section 2 satisfy any of the desirable properties highlighted in Section 2?
>
> See answer above.

---

> ### Author Response · Authors · 2018-11-14
> **Response to Review (pt. 1)**
>
> We thank the reviewer for their very knowledgeable review.  It gave us new insights, and made us see how our paper could be read in a way that we did not anticipate.
>
> In particular, we see how our writing may give the impression that we are trying to create a taxonomy of “prototype definitions.”  That was not at all our intention, as we elaborate on below.  Instead, inspired by the metric of Stock & Cisse (2018), we were interested in what were the differences between the examples contained in the dataset (both training and testing)---when evaluated by that metric, or the other four metrics we came up with---and how those differences might shed light on aspects such hard-to-learn and inherently-ambiguous submodes, memorized exceptions, and other concerns of example data corpus construction and curation.
>
> Below, we further respond to each of the reviewer’s comments:
>
> > Summary: This paper attempts to better understand the notion of prototypes and in some sense create a taxonomy for characterizing various prototypicality metrics. While the idea of thinking about such a taxonomy is novel, I think the paper falls in clearly justifying certain design choices such as why are the properties outlined at the beginning of Section 2 desirable. I also felt that the paper is resorting to rather informal ways of describing various properties and metrics without precisely quantifying them.
>
> We agree with the reviewer that it feels less-than-satisfactory to give such informal definitions for prototypes.  Indeed, this is how the list at the start of Section 2 came about: it is our own attempt at clarifying what we mean by “prototypes.” Before doing this work, we had no concrete definition for what “prototypes” actually were, whether they generally existed in data corpora for ML tasks, or---if they did---whether they corresponded to human intuition. Reading the existing literature on “prototypes” in the ML literature didn’t surface any precise definition: we found only informal statements and rather subjective goals for each metric, whereas the mechanism of each metric was often clearly defined.  Hence, for our own benefit, and that of the readers, we felt it was worth re-stating the common understanding from the prototype literature (even though it was vague); while doing this, we also added a few properties that seemed obvious, and were supported by our experiments, such as those of the last bullet in the list. However, we do not see this list as a real contribution of our work, and its removal would not affect our results.

---

### Official Review · AnonReviewer1 · 2018-11-13
**Unjustified heuristics, unclear if prototypes are useful, unconvincing experiments**

**Rating:** 3
**Confidence:** 3

**Review:**

Summary: The paper proposes methods for identifying prototypes. Unfortunately, a formal definition of a prototype is lacking, and the authors instead present a set of heuristics for sorting data points that purport to measure 'prototypicality', although different heuristics have different (and possibly conflicting) notions of what this means. The experiments are not very convincing, and often present results that are either inconclusive or negative, i.e. seem to demonstrate that prototypes are not very useful.

Pros:
- The notion of prototypes is used in various papers, but a formal definition is lacking, and the usefulness of prototypes is not demonstrated. The fact that this paper sets out to do both is laudable, although the paper needs work before it can be accepted for publications.

Detailed comments / cons:
*Defining prototypes:
  - The authors list desirable properties before defining (even informally) what a prototype is, and what its purposes are. Taking the first property as an example, is it reasonable to expect a metric for prototypes to be useful for image classification AND image generation? The answer completely depends on what one expects from a prototype, what its purpose is, etc.
  - The second property seems to indicate that prototypes are model-independent, i.e. two models trained on the same dataset will have the same prototypes. This is confusing as the metrics proposed are clearly model-dependent (e.g. adv completely depends on the trained model's decision boundary, conf obviously depends on the model providing the confidence score)
- The third and fourth property are poorly defined. Human intuition presupposes that humans agree on what a prototype means. Using 'modes of prototypical examples' in trying to define a metric for prototypes is circular, as a mode of prototypical example depends on a working notion of prototypical examples.
- The last property is completely dependent on which models are trained, and how they are trained. If a model has high label complexity, maybe it does not achieve high accuracy even when trained on high quality prototypes. In any case, this property is at odds with the first two properties.

In sum: it's not clear what prototypes are, so it becomes hard to judge if the list of desiderata is reasonable. The list is in any case ill-defined, and contains contradictions.

* Metrics for prototypicality
- The second paragraph in this section is unnecessary
- All of the metrics proposed are heuristics with little to no justification. Specific comments below.
- Adversarial robustness is a property of a trained model, not of prototypical examples, unless prototypes are supposed to be model dependent (contra property 1). In any case, it is not clear why examples that are robust to adversarial noise are good 'prototypes'.  Using facial recognition as an example, a 'mean face' may be very robust to adversarial noise but not prototypical at all under common definitions. A face with a particular type of facial hair (e.g. nose hair) may be very representative of a class of faces (i.e. a prototype), but very susceptible to adversarial noise. In fact, any examples in the boundary of the decision function will be more susceptible to adversaries, but that does not make them 'less prototypical'.
- Holdout retraining is again completely model dependent. Why should we expect a model to treat a prototype the same regardless of whether or not it is trained on it? This basically means that we expect the model to always be accurate on prototypes.
- Ensemble agreement proposes a notion of prototypes that is based on prediction 'hardness'. It is clear that such a notion depends completely on which models are being considered, which features are being used, and etc, much more than on notions of prototypicality inherent in the data. The same criticism applies to model confidence.
- Privacy preserving training assumes prototypicality has to do with the model being able to learn with some robustness to noise (related to Adversarial Robustness, but different). This assumes a definition of prototypes that is not congruent with the other metrics.

In sum: the proposed metrics are basically heuristics with little justification, and different metrics assume different notions of what a prototype is.

* Evaluation
- Section 3.1 claims that the metrics are strongly correlated, but that is not true for MNIST or CIFAR, and is somewhat true for fashion-mnist. In any case, since the metrics are so model-dependent, it is not clear if these results would hold if other models were used.
- Section 3.2 - The question asked of turkers in the study is too vague, and borderline irrelevant for the task at hand - what does the 'best image' of an airplane mean, and how does this translate to it being a prototype? All that the study demonstrates is that the proposed metrics score malformed images with low score. The results in Table 1 are very spread out, and seem to indicate a low agreement between the metrics and human evaluation - although Table 1 is almost irrelevant given the question that was asked of users.
- The results in Section 4 are very discouraging: sometimes it is better to train on most prototypical examples according to the metrics, sometimes it is worse, sometimes it's better to take examples in the middle. That is, prototypes don't seem to help at all. 'Prototype percentile' is uncorrelated with robustness for MNIST in Appendix E, while being correlated for other datasets. It is clear why this would be the case for metrics such as confidence, but in general models trained on less examples are less robust than models trained on the whole dataset (again, as expected). As a whole, the results do not provide any help for a user who wants to produce a more robust model, other than 'ignore prototypes and use the whole dataset'.

---

> ### Author Response · Authors · 2018-11-15
> **Response to Review (pt. 5)**
>
> * Evaluation
> > - Section 3.1 claims that the metrics are strongly correlated, but that is not true for MNIST or CIFAR, and is somewhat true for fashion-mnist. In any case, since the metrics are so model-dependent, it is not clear if these results would hold if other models were used.
>
> Almost all pairs have a correlation coefficient higher than .5, and many have correlation coefficients higher than .7. Again, for the reasons stated above, our metrics are not model-dependent.
>
> > - Section 3.2 - The question asked of turkers in the study is too vague, and borderline irrelevant for the task at hand - what does the 'best image' of an airplane mean, and how does this translate to it being a prototype? All that the study demonstrates is that the proposed metrics score malformed images with low score. The results in Table 1 are very spread out, and seem to indicate a low agreement between the metrics and human evaluation - although Table 1 is almost irrelevant given the question that was asked of users.
>
> We completely reject the reviewer’s statements here. The results show a very strong correlation with human intuition; indeed, the correlation with the top decile in the “pick best” table is quite unexpectedly strong for some metrics.  The correlation varies with metrics, and for some metrics the strongest correlation is indeed on finding the non-prototypes (i.e., outliers), but this can only be expected for datasets where the majority of examples seem “good” to humans.
>
> Regarding the study design, it follows best practice in evaluating whether human intuition of what are “good” examples for a class matches our metrics’ rankings. The question must be generic: if we taught humans what a prototype is based on the properties we used to design our metrics, this would result in a circular argument and unsound results. Our experiments with Turkers instead show that our metrics identify examples in the training and test data that are consistent with what a human would intuitively consider as a “good example” without priming them with what we consider as a good example.
>
> > - The results in Section 4 are very discouraging: sometimes it is better to train on most prototypical examples according to the metrics, sometimes it is worse, sometimes it's better to take examples in the middle. That is, prototypes don't seem to help at all. 'Prototype percentile' is uncorrelated with robustness for MNIST in Appendix E, while being correlated for other datasets. It is clear why this would be the case for metrics such as confidence, but in general models trained on less examples are less robust than models trained on the whole dataset (again, as expected). As a whole, the results do not provide any help for a user who wants to produce a more robust model, other than 'ignore prototypes and use the whole dataset'.
>
> As explained in the text of our paper, the results are not discouraging but rather show that the metrics are able to capture the subtleties of different datasets. MNIST is a simple task where almost all training points are correctly classified and therefore models trained on this dataset learn more from outlier examples. Instead, FashionMNIST is a more complex task where some points are either mislabeled or ambiguous (it is unclear to which class they belong): therefore models trained on the least prototypical examples do not perform well (because these training points are mislabeled) but models trained on the next slices of data according to the prototypicality metric perform in a similar way to the MNIST dataset.
>
> Furthermore, the truth is rarely black-and-white. While much prior work has either argued it is better to train on the easy prototypical examples, or conversely argued it is better to train on the harder (more outlier) examples, we find that both statements can be true depending on the exact details of the metric used, dataset, or learning task. We do not find our results to be discouraging, but even if they were, a true but discouraging result is worth reporting even when it does not match one’s expectations.

---

> > ### Comment · AnonReviewer1 · 2018-11-15
> > **Brief response pt.1**
> >
> > -> If prototypes exist, why would they not exist for both classification and generation tasks?  Further, if they do, certainly it might be desirable if a single technique could find the prototype for both tasks. The reviewer rejects that this may be desirable, or at least implies that its desirability is unreasonable. As we highlight in the introduction, an independent contribution of our work is the discovery that the adversarial metric is highly-correlated with the retraining metric. This means that one could find prototypes using the retraining metric when it is difficult to use the adversarial metric (e.g., on word embeddings). This is desirable, or at least reasonably so.
> >
> > Note that I did not reject the desirability of general techniques, rather I pointed out that it is impossible to judge if this is reasonable or not without specifying what prototypes are, and what they are for. Prototype is an abstract concept, and as such a question that begins with ‘if prototypes exist’ is nonsensical without a definition of what a prototype is.
> >
> > -> Again, the reviewer objects to this characteristic being listed as desirable. Even if it was an impossible goal, we fail to see why it would be so objectionable to list it as desirable. However, in this case, the reviewer’s emphatic complaints are not just unsupported, they are directly contradicted by the literature on adversarial examples.  Adversarial examples transfer from one model to another, which suggests that the adversarial distance metric will be in large parts independent of the model architecture. Furthermore, when computed over an ensemble of models (rather than a single model), the confidence metric we provide is also empirically stable with regard to model architecture.
> >
> > Again, I am not rejecting this characteristic as desirable. My criticism is aimed at defining a property as desirable for prototypes and then proposing a set of prototype metrics that directly contradict the property. It is absolutely not clear from the text if prototypes are a property of the dataset / task or a property of a (dataset, trained-model) pair, as this property seems to indicate that it is the former while the metrics indicate it is the latter.
> >
> > -> Correspondence to human intuition has been the key defining feature of prototypes in all earlier work that we found in the ML literature. This is why we include it as a desirable property here. We do not presuppose anything about humans; instead, we test whether their impressions agree with our metrics.  Our experiments conclusively show that human intuition agrees strongly with some of our metrics, and with all of our metrics less strongly.
> >
> > Again, the authors need to recognize that we are dealing with an abstract concept. The fact that some of the proposed metrics are correlated (I reject that the correlation is even very strong, see below) with human’s response to the question ‘What is the best / worst image of X’ does NOT conclusively show that the metric aligns with human intuition of what prototypes are, UNLESS prototypes are defined as ‘the best / worst images of X’.

---

> > > ### Comment · AnonReviewer1 · 2018-11-15
> > > **pt.2**
> > >
> > >
> > > -> This statement by the reviewer is either confused, or saying something completely unsupported. This last property is certainly not at odds with the first two desirable properties, and this last property has been validated by our experiments (see Sections 3 and 4 and the new, expanded text of the caption in Section 4).
> > >
> > > The first property states that metrics should be independent of the learning task, and the second that they should be independent of the modeling approach.
> > > The last property states that models trained on prototypes should have good test accuracy.
> > > Test accuracy is a property of certain learning tasks (e.g. classification), but not of others (e.g. unsupervised learning, text generation, clustering), so it is clear that the last property conflicts with the first.
> > > How a model performs when you restrict its training set is clearly model-dependent, the obvious example being a severely regularized model which would work well with lots of data but poorly with a small amount of data. Thus, the last property conflicts with the second. Whether or not the models used in the experiments display this behavior is irrelevant - the point is that the last property depends on which model is used.
> > >
> > >
> > > -> Holdout retraining may be model independent because an outlier (e.g., a point close to the decision boundary) may be easier to forget for a model than a prototype (e.g., a point at the center of a dense class mode).  Again, we make no a-priori claim; instead, we verify the benefit of this prototypicality metric in after-the-fact experiments, and find that it is highly correlated with the other metrics (and especially highly correlated with ‘adv’).
> > > -> Because a large number of models are used in the ensemble, and the ensemble and confidence metric measure the consensus among these models, the resulting metrics do not depend on the specific models used in the ensemble. Again, we make no a-priori claim; instead, we verify the benefit of this prototypicality metric in after-the-fact experiments.
> > >
> > >
> > > It is clear that the authors are operating under a notion of model independency that is alien to this reviewer. If something depends on what model or class of models is used, it is a misnomer to call it model independent. The experiments presented do not touch on this, as they rely on only one model being trained on multiple datasets.
> > >
> > >
> > > -> The reviewer seems led astray here by their implicit, unstated assumption about what they feel to be a “correct” notion of prototypes.  In after-the-fact experimental evaluation, each of our metrics comports well to our list of desirable properties for prototypes.  We do not try to provide strong a-priori justifications, since intuitions are often misleading (see the paragraph which the reviewer deemed superfluous).  Furthermore, a main contribution of our work is to show how the very differences between the metrics can tell us a lot about the training and test data.
> > >
> > > Iit is important for the authors to note that since there is no clear task presented in this paper, they cannot just say ‘we showed that this worked in experiments’, as someone could in a well defined task where accuracy is improved. The question asked to measure human intuition is ‘Select the Best / Worst image of CLASS_NAME’. If the authors think that finding metrics that correlate with human answers on this question for a particular model is useful, they have to argue for it. The experiments that could be of any practical use all have either inconclusive or bad results. The authors try a myriad of things - some help sometimes, while hurting at other times - there is no clear conclusion, and no practical advice for practitioners.

---

> ### Author Response · Authors · 2018-11-15
> **Response to Review (pt. 4)**
>
> > Using facial recognition as an example, a 'mean face' may be very robust to adversarial noise but not prototypical at all under common definitions. A face with a particular type of facial hair (e.g. nose hair) may be very representative of a class of faces (i.e. a prototype), but very susceptible to adversarial noise. In fact, any examples in the boundary of the decision function will be more susceptible to adversaries, but that does not make them 'less prototypical'.
>
> We don’t understand these objections by the reviewer. A “mean face” would not exist in the training or test data, and hence not be subject to our metric. Also, we cannot speculate on the reviewer’s intuition about facial images with nose hair. We can only point out that the “adv” metric successfully finds prototypical examples in experiments, successfully meets our desirable properties, and is clearly useful for the purposes of curriculum learning.  Perhaps the reviewer could state concretely what other goals they believe a ‘prototype’ metric should meet.
>
> > - Holdout retraining is again completely model dependent. Why should we expect a model to treat a prototype the same regardless of whether or not it is trained on it? This basically means that we expect the model to always be accurate on prototypes.
>
> Holdout retraining may be model independent because an outlier (e.g., a point close to the decision boundary) may be easier to forget for a model than a prototype (e.g., a point at the center of a dense class mode).  Again, we make no a-priori claim; instead, we verify the benefit of this prototypicality metric in after-the-fact experiments, and find that it is highly correlated with the other metrics (and especially highly correlated with ‘adv’).
>
> > - Ensemble agreement proposes a notion of prototypes that is based on prediction 'hardness'. It is clear that such a notion depends completely on which models are being considered, which features are being used, and etc, much more than on notions of prototypicality inherent in the data. The same criticism applies to model confidence.
>
> Because a large number of models are used in the ensemble, and the ensemble and confidence metric measure the consensus among these models, the resulting metrics do not depend on the specific models used in the ensemble. Again, we make no a-priori claim; instead, we verify the benefit of this prototypicality metric in after-the-fact experiments.
>
> > - Privacy preserving training assumes prototypicality has to do with the model being able to learn with some robustness to noise (related to Adversarial Robustness, but different). This assumes a definition of prototypes that is not congruent with the other metrics.
>
> Adversarial distance (our “adv” metric) does not involve changing the training procedure at all, and are also not based on noise (rather, on directed search).. However, privacy-preserving learning algorithms will---by design---fail to learn about outliers and data that is found very rarely in the training data. Thus, by varying the level of privacy, it should be possible to rank examples from those that are easiest to learn, and at the heart of the learned distribution, to those that are hardest to learn and at the distribution’s boundary. This is directly related to our other metrics, not incongruous. But, just as with the other metrics, we make no a-priori claim; instead, we verify the benefit of this prototypicality metric in after-the-fact experiments.
>
> > In sum: the proposed metrics are basically heuristics with little justification, and different metrics assume different notions of what a prototype is.
>
> The reviewer seems led astray here by their implicit, unstated assumption about what they feel to be a “correct” notion of prototypes.  In after-the-fact experimental evaluation, each of our metrics comports well to our list of desirable properties for prototypes.  We do not try to provide strong a-priori justifications, since intuitions are often misleading (see the paragraph which the reviewer deemed superfluous).  Furthermore, a main contribution of our work is to show how the very differences between the metrics can tell us a lot about the training and test data.

---

> ### Author Response · Authors · 2018-11-15
> **Response to Review (pt. 3)**
>
> > - The third and fourth property are poorly defined. Human intuition presupposes that humans agree on what a prototype means.
>
> Correspondence to human intuition has been the key defining feature of prototypes in all earlier work that we found in the ML literature. This is why we include it as a desirable property here. We do not presuppose anything about humans; instead, we test whether their impressions agree with our metrics.  Our experiments conclusively show that human intuition agrees strongly with some of our metrics, and with all of our metrics less strongly.
>
> > Using 'modes of prototypical examples' in trying to define a metric for prototypes is circular, as a mode of prototypical example depends on a working notion of prototypical examples.
>
> The circularity is noted. The word prototype should be removed from the body of this bullet.
>
> > - The last property is completely dependent on which models are trained, and how they are trained. If a model has high label complexity, maybe it does not achieve high accuracy even when trained on high quality prototypes. In any case, this property is at odds with the first two properties.
>
> This statement by the reviewer is either confused, or saying something completely unsupported. This last property is certainly not at odds with the first two desirable properties, and this last property has been validated by our experiments (see Sections 3 and 4 and the new, expanded text of the caption in Section 4).
>
> > In sum: it's not clear what prototypes are, so it becomes hard to judge if the list of desiderata is reasonable. The list is in any case ill-defined, and contains contradictions.
>
> From our rebuttal, it should be clear to an objective reader that there are no contradictions between the different properties we specified in Section 2.  We have pointed to the relevant sections of our paper demonstrating that the list of desiderata is reasonable.
>
> * Metrics for prototypicality
> > - The second paragraph in this section is unnecessary
>
> This paragraph describes approaches that we tried, but which failed.  It is included because the presentation of negative results is an important part of the scientific method, and we felt it should be included to provide a complete and honest description of our work.
>
> > - All of the metrics proposed are heuristics with little to no justification. Specific comments below.
>
> The techniques are justified *after-the-fact* by their empirical validation: each technique does indeed find prototypical examples (and outliers), corresponding to the properties that we outlined as being desirable in Section 2.  By objecting to the lack of a-priori justification, and by using phrasing such as “good ‘prototypes’” and “not prototypical at all under common definitions,” the reviewer seems to be looking for a reason why these metrics would match some notion of prototypicality that the reviewer has in mind---but which the reviewer is not stating.
>
> > - Adversarial robustness is a property of a trained model, not of prototypical examples, unless prototypes are supposed to be model dependent (contra property 1). In any case, it is not clear why examples that are robust to adversarial noise are good 'prototypes'.
>
> We repeat here that we do not measure adversarial robustness but rather the distance of test inputs to the decision boundary (which can be approximated using adversarial example techniques). Given that adversarial examples transfer across model architectures, the distance to the decision boundary will be reasonably independent of the model. Empirically, we trained multiple models with different architectures and found that using distance to the decision boundary as determined by an adversarial example attack was consistent.
>
> With this metric, and the remaining metrics, we agree that, a-priori, it is not obvious that they should find prototypes.  However, as we demonstrate in our experiments (for the “adv” metric, validating the earlier results of [Stock and Cisse, 2018]), each metric does indeed rank examples in a manner that comports with the common understanding and usage of prototypes.

---

> ### Author Response · Authors · 2018-11-15
> **Response to Review (pt. 2)**
>
> Pros:
> > - The notion of prototypes is used in various papers, but a formal definition is lacking, and the usefulness of prototypes is not demonstrated. The fact that this paper sets out to do both is laudable, although the paper needs work before it can be accepted for publications.
>
> Our paper does not try to provide a formal definition of prototypicality. Our intent is to provide different metrics for quantifying properties that help to identify “prototypes” and more importantly demonstrate how these metrics can be combined to help understand the training and test data (e.g., find memorized exceptions or mislabeled and inherently-ambiguous training data) as well as other aspects of learning algorithms (e.g., curriculum learning) . Before doing this work, we had no concrete definition for what “prototypes” actually were, whether they generally existed in data corpora for ML tasks, or---if they did---whether they corresponded to human intuition.
>
> Reading the existing literature on “prototypes” in the ML literature didn’t surface any precise definition: we found only informal statements and rather subjective goals for each metric, whereas the mechanism of each metric was often clearly defined.  Hence, for our own benefit, and that of the readers, we felt it was worth re-stating the common understanding from the prototype literature (even though it was vague); while doing this, we also added a few properties that seemed obvious, and were supported by our experiments, such as those of the last bullet in the list. However, we do not see this list as a real contribution of our work, and its removal would not affect our results.
>
> >Detailed comments / cons:
> >*Defining prototypes:
>
> Prototypes have been defined implicitly and informally in previous work, but in all the previous definitions the common thread has been that (a) prototypes are a smaller set of examples that characterizes well the learning task, with good coverage, and (b) the set of examples deemed to be prototypes agree with human intuition.  Our desirable properties simply capture and expand on that well-established definition. It is clear that this reviewer may not agree with that definition; however, they offer no alternative, and we believe none can be found in the ML literature.
>
> >  - The authors list desirable properties before defining (even informally) what a prototype is, and what its purposes are. Taking the first property as an example, is it reasonable to expect a metric for prototypes to be useful for image classification AND image generation? The answer completely depends on what one expects from a prototype, what its purpose is, etc.
>
> If prototypes exist, why would they not exist for both classification and generation tasks?  Further, if they do, certainly it might be desirable if a single technique could find the prototype for both tasks. The reviewer rejects that this may be desirable, or at least implies that its desirability is unreasonable. As we highlight in the introduction, an independent contribution of our work is the discovery that the adversarial metric is highly-correlated with the retraining metric. This means that one could find prototypes using the retraining metric when it is difficult to use the adversarial metric (e.g., on word embeddings). This is desirable, or at least reasonably so.
>
> >   - The second property seems to indicate that prototypes are model-independent, i.e. two models trained on the same dataset will have the same prototypes. This is confusing as the metrics proposed are clearly model-dependent (e.g. adv completely depends on the trained model's decision boundary, conf obviously depends on the model providing the confidence score)
>
> Again, the reviewer objects to this characteristic being listed as desirable. Even if it was an impossible goal, we fail to see why it would be so objectionable to list it as desirable. However, in this case, the reviewer’s emphatic complaints are not just unsupported, they are directly contradicted by the literature on adversarial examples.  Adversarial examples transfer from one model to another, which suggests that the adversarial distance metric will be in large parts independent of the model architecture. Furthermore, when computed over an ensemble of models (rather than a single model), the confidence metric we provide is also empirically stable with regard to model architecture.

---

> ### Author Response · Authors · 2018-11-15
> **Response to Review (pt. 1)**
>
> We thank the reviewer for taking the time to read our paper in detail, and for providing such extensive comments. Unfortunately, it appears that the reviewer has a specific definition of “prototypes” in mind, which seems fundamentally incompatible with our investigation and results. Since the reviewer doesn’t explicitly state what they believe to be a “correct” definition, and since our definition is much the same as that in prior work (albeit more detailed), we find the reviewers forceful objections to our work hard to comprehend.
>
> We respond to each of the reviewer’s points below:
>
> > Review: Summary: The paper proposes methods for identifying prototypes. Unfortunately, a formal definition of a prototype is lacking, and the authors instead present a set of heuristics for sorting data points that purport to measure 'prototypicality', although different heuristics have different (and possibly conflicting) notions of what this means. The experiments are not very convincing, and often present results that are either inconclusive or negative, i.e. seem to demonstrate that prototypes are not very useful.
>
> Our paper provides a definition of what constitutes a “prototype” that is more clear than those used in previous work. Below, the reviewer says that this is laudable, while above they object to the unfortunate informality of our definitions and label our technical metrics as “heuristics” and our goals as being “purported”.  As we explain below, and in our response to AnonReviewer2, there is little reason to believe the notion of prototypes lends itself to precise, formal definitions without a concrete basis in quantitative metrics.
>
> As for our mechanisms, they no more heuristics than most techniques in machine learning. As for our goals, they are more (not less) clear than those in prior work, and our experiments in curriculum learning and our human evaluation demonstrate that we achieve those goals.
>
> Finally, the reviewer emphatically finds fault with a number of our definitions, the conduct of our research, and the results of our experiments. In particular, the reviewer makes many statements where they reject the conclusions or statements (sometimes mis-characterized) from our work. The reviewer’s dismiss our work with great confidence, often making countervailing statements  of their own without offering any support. We cannot explain the reviewer’s strong objections. However, we must explain where their position is confused, unfounded, or simply incorrect.

---

### Meta-Review · Area_Chair1 · 2018-12-16
**Make definitions more precise**

**Confidence:** 4
**Recommendation:** Reject

**Metareview:**

This paper considers "prototypes" in machine learning, in which a small subset of a dataset is selected as representative of the behavior of the models. The authors propose a number of desiderata, and outline the connections to existing approaches. Further, they carry out evaluation with user studies to compare them with human intuition, and empirical experiments to compare them to each other. The reviewers agreed that the search for more concrete definitions of prototypes is a worthy one, and they appreciated the user studies.

The reviewers and AC note the following potential weaknesses: (1) the specific description of prototypes that the authors are using is not provided precisely, (2) the desiderata was found to be informal, leading to considerable confusion regarding the choices that are made and their compatibility with each other, (3) concerns in the evaluation regarding the practicality and the appropriateness of the user study for the goals of the paper.

Although the authors provided detailed responses to these concerns, most of them still remained. Both reviewer 1 and reviewer 2 encourage the authors to define the prototypes defined more precisely, providing motivation for the various choices therein. Even though some of the concerns raised by reviewer 3 were addressed, it still remains to be seen how scalable the approach is for real-world applications.

For these reasons, the reviewers and the AC feel that the authors would need to make substantial improvements for the paper to be accepted.